# Jamming with magnetic composites

Buse Aktaş ®[1,2] ✉, Minsoo Kim ®[1] ✉, Marc Bäckert[1], Gianluca Sicilia[1], Gian-Luca Franchini[1], Florian Heemeyer[1], Simone Gervasoni[1], Xiang-Zhong Chen[3,4], Salvador Pané ®[1] & Bradley J. Nelson ®[1]

The jamming transition—marked by dramatic changes in mechanical properties, such as stiffness and damping—enables programmable and adaptive structures for robotic applications. This phenomenon, driven by changes in the coupling between individual subunits of an aggregate, can be controlled through external actuation sources. Existing jamming actuation methods, such as applying a vacuum with an airtight envelope, pose significant limitations, as they require the structures to be tethered, limiting reconfigurability and scalability. Here, we introduce an untethered jamming mechanism based on magnetic interactions between soft-ferromagnetic composites. We establish composite design principles to program the magnetization of the subunits, demonstrate linear, planar, and volumetric jamming and shape-locking, and model the magneto-mechanical behavior. This approach contributes to the development of jamming-based materials in which the jamming directions and transition points can be tuned on-the-fly by adjusting the external magnetic field orientation and strength, respectively.

Jamming is a phenomenon in which an aggregate transitions from a fluid-like state to a solid-like state when the kinematic and frictional coupling between the individual parts change[1–3]. Some jamming transitions are undesired, such as traffic jams on highway exits or clogs of granules in funnels. If the jamming transition can be purposely controlled, the resulting changes in bulk mechanical properties can be advantageous for engineering applications[4,5]. Programmable mechanical tunability enables material systems to adapt to changes in their environment by not only having strength and precision in their stiff state and being gentle and conformable in their soft state, but also by actively modifying their energy dissipation performance (damping). This tunability is particularly useful for applications such as surgical, biomedical, and wearable robots, where the environment is both delicate and dynamic[6–8]. Existing jamming actuation methods, such as vacuum and voltage, are not compatible with robotic tasks in hard-to-reach areas, such as constricted areas inside the human body, since they require the structure to be tethered (via wiring or tubing) or require bulky on-board actuation mechanisms (pumps and/or batteries)[9–11].

Here, we present an untethered jamming mechanism utilizing remote magnetic actuation. We create magnetic-jamming structures with magnetic composite subunits containing soft-ferromagnetic materials. The application of external magnetic fields induces magnetic forces between the individual subunits to enable their remote jamming and unjamming. Analytical and numerical studies outline design rules for the magnetic composite subunits to predict, program, and actively control the mechanical properties of the resulting jamming structures (e.g., stiffness and yield). Finally, we demonstrate multi-dimensional jamming in 1-D to 3-D structures and introduce the ability to selectively jam structures along specific directions. We anticipate this will help establish a class of assembly-based magneto-mechanical structures with multi-dimensional and spatially varying tunability.

## Results

### Soft-ferromagnetic composites for jamming

The proposed jamming structure is made of individual composite subunits that consist of soft-ferromagnetic elements within a non-

[1]Multi-Scale Robotics Laboratory, Institute of Robotics and Intelligent Systems, ETH Zurich, Zurich, Switzerland. [2]Robotic Composites and Compositions Group, Max Planck Institute for Intelligent Systems, Stuttgart, Germany. [3]International Institute for Intelligent Nanorobots and Nanosystems, College of Intelligent Robotics and Advanced Manufacturing, State Key Laboratory of Photovoltaic Science and Technology, Fudan University, Shanghai, People's Republic of China. [4]Yiwu Research Institute of Fudan University, Yiwu, China. ✉e-mail: buse.aktas@is.mpg.de; minkim@ethz.ch

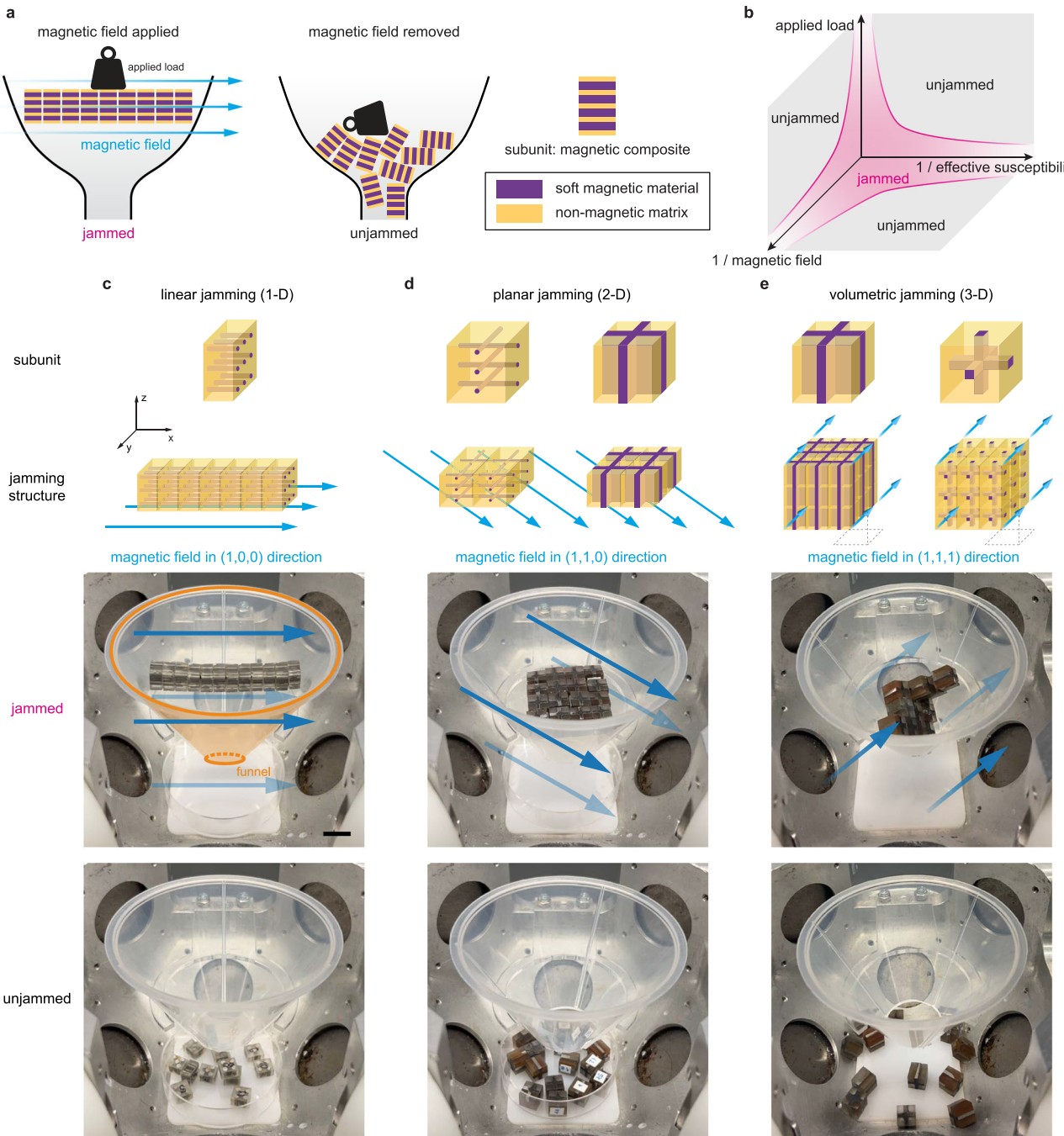

**Fig. 1 | Magnetically-induced jamming. a** Magnetic composite design enables jamming that can be actuated with an externally applied magnetic field. **b** The phase of the resulting structure depends on the applied load, the applied external magnetic field, and the effective susceptibility of the individual subunits (material's magnetic properties, volume, and distribution). **c**–**e** Funnel experiments demonstrate how multi-dimensional magnetic jamming can be achieved in one, two, and three dimensions through composite design. The structures can withstand gravitational forces and maintain structural integrity when an external magnetic field is applied, while they collapse entirely once the field is removed (Scale bar: 1 cm). See Supplementary Movie 1. While these experiments focus on illustrating the transition from jammed to unjammed, Supplementary Movie 2 illustrates the transition from unjammed to jammed.

magnetic matrix. In the composite, the magnetic elements are aligned with the desired direction of jamming and are continuous along that direction. An external field is then applied along that direction, and each subunit becomes magnetized, and the attractive forces between neighboring subunits result in jamming. Figure 1a demonstrates this behavior in one dimension. When a magnetic field is present, the chain of subunits acts like a continuous beam, representing a jammed structure. When the field is removed, there is no (or minimal) attractive force remaining, and the subunits act as independent constituents.

Here, the morphology of the structure and the composite design of the subunits ensure that magnetically induced jamming behavior arises primarily from magnetic attraction forces. This allows for a direct observation of magnetic jamming, independent of kinematic interlocking. In contrast, previous studies have observed jamming in magnetic particle aggregates, where homogeneous particles rearrange in response to the applied magnetic field to reach energetically favorable positions, while magnetically jamming in the direction of the applied magnetic field[12,13].

A phase diagram for magnetization-based jamming structures can be drawn based on prior work on jamming, which typically consists of a three-dimensional phase diagram with dependencies on density, temperature, and applied load[1,14]. The adapted three-dimensional phase diagram shown in Fig. 1b has three axes. These axes are determined by the applied external load, the applied magnetic field, and the effective susceptibility of the composite subunit. The shape of the phase diagram affirms prior research on the jamming of attractive particles, where the attraction was controlled by the addition of a dispersant[15]. A quantitative version of the diagram is shown in Supplementary Fig. 1. The jammed state is in the center of this Cartesian space, where an increase in the applied magnetic field can jam a system. Typical to any jamming system, an increase in the applied load unjams the structure. The effective susceptibility of the subunits ($\chi_{\text{eff}}$) can be determined experimentally or numerically by $\mathbf{M} = \chi_{\text{eff}}\mathbf{H}$, where $\mathbf{M}$ is the resulting magnetization of the subunit, and $\mathbf{H}$ is the applied external field[16]. This value depends on the overall size and shape of the subunit, the total magnetic volume within the composite (or the volume fraction), the distribution and alignment of the magnetic material within the composite, and the magnetic properties of the material. The effective susceptibility can be pre-programmed with the composite design, as described in the next section of this article.

Soft-ferromagnetic materials that have low remanence and low coercivity were selected to ensure reversible magnetic-jamming behavior. Low remanence allows low magnetic moments when there is no external magnetic field, introducing the possibility of unjamming the structure. Low coercivity allows for easy magnetization and demagnetization (and as a result, easy jamming and unjamming) using relatively low fields.

Additionally, the shape-dependent magnetization behavior of soft-ferromagnetic materials was leveraged for designing subunits. Specifically, a continuous soft-magnetic body has an "easy axis", a direction along which it is more energetically favorable to magnetize, typically the more elongated dimension of the body[17,18]. If a jamming structure's subunits are made of a continuous soft-ferromagnetic material, jamming will preferably occur along the longer axis of the subunit. This effect will be amplified when the subunits are not spatially constricted, as they will also rotate to align their easy axes along the magnetic field direction. This limits the possible morphologies and functionalities of the resulting jamming structures.

Magnetic composite subunits that have soft-ferromagnetic components within a non-magnetic matrix can help overcome these limitations. The magnetic constituents' easy axis can be aligned along the desired direction of jamming irrespective of the overall geometry of the subunits, enabling programmability of jamming directions. Moreover, multiple discrete easy axes can be pre-programmed into a subunit, enabling multi-degree-of-freedom control, which is the focus of the section on multi-directional jamming.

Jamming structures with different subunit geometries satisfy different design specifications and can be leveraged to achieve specific mechanical behaviors[2,19]. For example, in layer jamming (used in robotic structures, such as wearables, to achieve dramatic changes in bending stiffness[20]), the behavior is dominated by the frictional forces between the layers. This type of jamming cannot be achieved with continuous layers of soft-ferromagnetic materials, since a free-standing homogeneous sheet cannot typically be magnetized along its shortest dimension. However, as shown in Fig. 1c, composite layers in which pillar-like magnetic structures are oriented along the shortest dimension (perpendicular to the layer's main plane) can enable layer jamming.

Another example is fiber jamming, which requires force chains in two dimensions along the directions orthogonal to the main axis of the fibers. Whereas granular jamming benefits from forces along all three dimensions[19]. Figure 1d, e demonstrates how the concept from one dimension can be extended to two and three dimensions by creating elongated magnetic features in all directions where jamming is desired. This results in multiple easy axes orthogonal to each other that are all equally magnetized when a magnetic field is applied along a direction between these axes at a 45-degree angle. This enables jamming in more than one dimension, which is not possible with a continuous magnetic material. This versatility, enabled by composite design, introduces possibilities for soft-magnetic assemblies and swarms[21–23].

## Magnetization of the composite

The magnetic hysteresis behavior of the soft-ferromagnetic material used is shown in Fig. 2a. The operating region, based on the desired range of magnetic fields applicable, lies in the linear region of the hysteresis loop, far from saturation. This allows a linear approximation with zero remanence, zero coercivity, and a constant susceptibility, where the magnetization linearly increases with the applied magnetic field. The increase in magnetization results in an increase in the jamming force between two neighboring composite subunits, which in turn results in a remotely tunable jamming behavior (Fig. 2b).

The magnetic moment ($\mathbf{m}$) of a single homogeneous magnetic body in the linear region of the hysteresis loop can be defined by the equation: $\mathbf{m} = \mathbf{H}V\chi$, where $\mathbf{H}$ is the applied magnetic field, $V$ is the volume of the magnetic body, and $\chi$ is the magnetic susceptibility tensor that depends on the shape and the material's magnetic properties. Within the proposed composite, multiple magnetic bodies neighbor each other, and each body creates its own magnetic field once magnetized. Neighboring bodies demagnetize each other[24]. For example, when two pillars are next to each other and are magnetized along their longer axis with the same magnetic field (Fig. 2c), the magnetization of one of the pillars can be shown with a modified version of the previous equation: $m = \frac{H}{\frac{1}{V\chi} + \frac{1}{4\pi s^3}}$, where $V$ is the volume of a pillar, $\chi$ is the component of susceptibility of the pillar along its main axis (which depends on the pillar's aspect ratio and the material's inherent susceptibility), and $s$ is the distance between the two pillars. This analytical model, which approximates the pillars as magnetic dipoles at a single point, was compared with simulations and experiments. The results demonstrate and confirm two main trends to consider when designing magnetic composites given a constant total magnetic volume: (1) magnetic elements with a higher aspect ratio within the composite will result in higher magnetization (Fig. 2d), and (2) if the magnetic elements are tightly packed, they will exhibit lower magnetization due to internal demagnetization effects. Moreover, at a specific spacing $s$, the distance between pillars is sufficient to ensure they will not influence each other's magnetization (when $4\pi s^3 \gg V\chi$) (Fig. 2e).

For a given application, design constraints pre-determine the overall size and geometry of a subunit within a jamming structure. Within those limitations, there are two remaining design parameters to consider: how much magnetic material to use within the total volume, and how to distribute the magnetic material within that volume. The total magnetic volume can be distributed into thinner but more numerous pillars within the subunit, leveraging the aspect ratio effect for increased magnetization. Since there is a predetermined volume for the overall subunit, as the number of pillars increases, the distance between the pillars is reduced, increasing the demagnetization effect. The validated two-pillar finite element simulations from above were extended to a two-dimensional grid of pillars within a constant volume composite to analyze these relationships and trade-offs. The effective susceptibilities of the composites were extracted from simulations (Fig. 2f, g). The results demonstrate that: (1) the aspect ratio of the overall subunit dictates the maximum achievable effective susceptibility, (2) an increase in the number of pillars per volume increases the effective susceptibility, yet this effect plateaus as the pillar spacing reduces, and (3) increasing the amount of magnetic volume within the

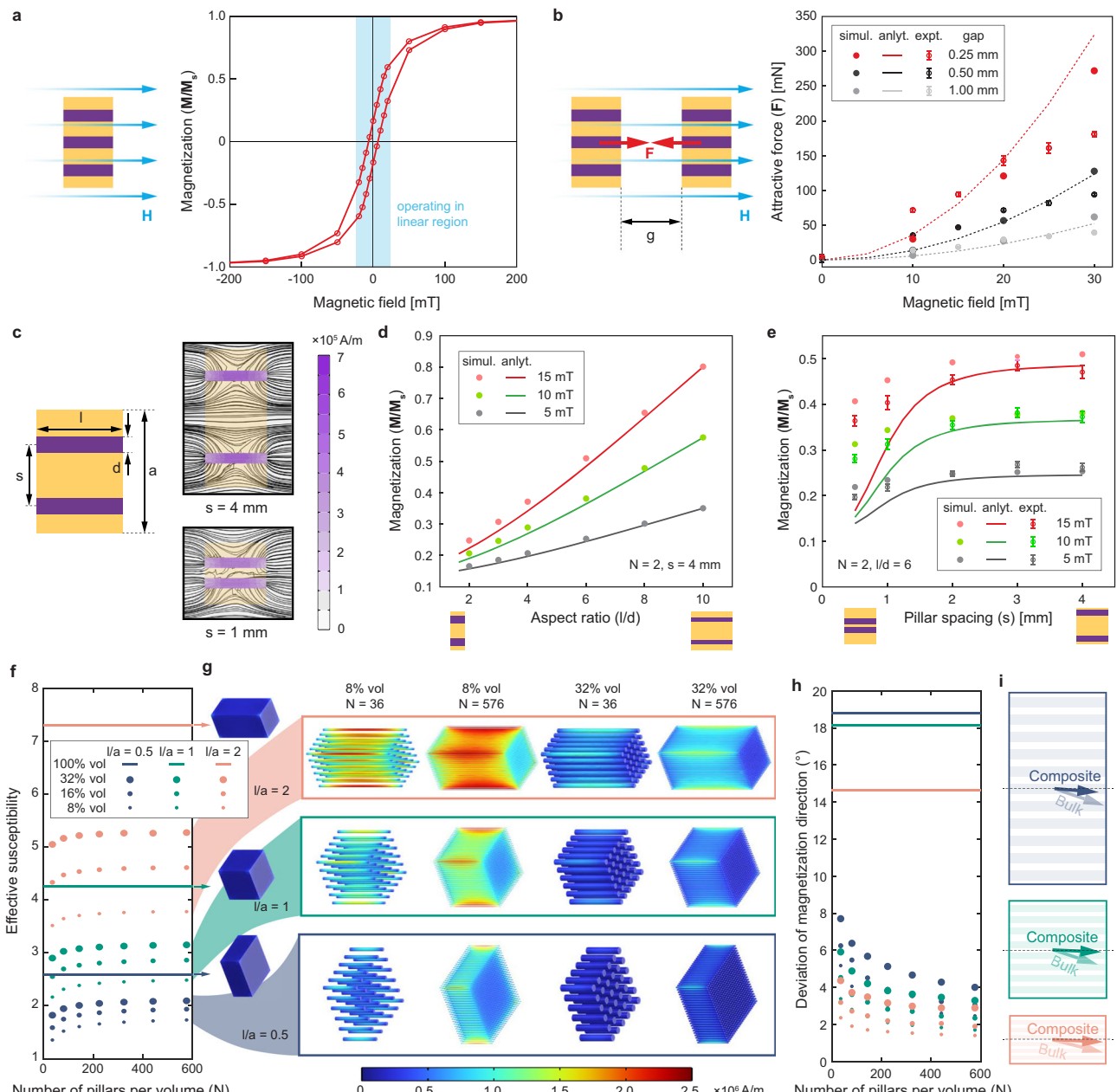

**Fig. 2 | Magnetization behavior of one-dimensional composites. a** Hysteresis loop of soft-ferromagnetic material illustrates operation in the linear region. **b** Simulations and experiments show that an increase in the magnetic field results in an increase in the attractive force between subunits. **c** Geometric entities in a composite fundamentally influence magnetization performance **d** Given constant magnetic volume, increased aspect ratio improves magnetization. **e** Larger pillar spacing results in higher magnetization until a constant is reached. **f** Given a constant total volume of a subunit, its effective susceptibility is investigated as a function of: the aspect ratio of the subunit, total magnetic volume within the subunit, and the number of magnetic pillars per volume. **g** Simulations demonstrate that pillars within a composite demagnetize each other. **h** The magnetization direction's deviation from the desired jamming direction is reduced with more pillars per volume. **i** The magnetization directions are illustrated as vectors. Error bars show standard error.

composite has a greater influence for (a) a lower number of pillars per volume and (b) subunits with higher aspect ratios.

While the effective susceptibility of a composite can never reach the susceptibility of an equivalent continuous magnetic material (as can be verified in Fig. 2f), creating a composite is essential to be able to magnetize free-standing subunits in a desired direction, enabling programmable jamming directions. A continuous square prism fixed in space and magnetized along its longer dimension can have a slight deviation in the direction of magnetization, as the direction of magnetization can be between its easy axes (along the diagonals) and the

applied magnetic field. A composite made of individual pillars along one direction, on the other hand, can have a much better aligned magnetization direction. A quantitative comparison of the ability of the composites to direct the magnetization relative to a continuous magnetic structure is presented in Fig. 2h, i. The impact is especially significant when the aspect ratio of the subunit ($l/a$) is smaller and the aspect ratio of the individual pillars is higher ($l/d$). By enabling control over the direction of magnetization, magnetic composites enable programmability of the directions of the attraction forces between subunits within a jamming structure.

## Magnetically induced mechanical behavior

The mechanical behavior of a jamming-based system is determined by the force chains formed throughout the structure along contact points or surfaces between neighboring subunits[25]. In purely mechanical jamming structures, this is induced by external stresses applied due to either a constraining geometry (e.g., a funnel) or an applied pressure (e.g., an airtight casing connected to a vacuum source). In magnetically induced jamming, the behavior is magneto-mechanical, i.e., the force chains are formed primarily due to the attraction between neighboring subunits. There are three possible interaction modes between neighboring subunits as they are perturbed by an external load: sliding, pivoting, and separating. The mode that takes precedence depends not only on the inter-subunit forces, but also on how the external force applied propagates within the structure. These modes, which are the building blocks of the force chains, are investigated both analytically and experimentally, demonstrating the functional programmable mechanical behaviors that magnetically induced jamming structures can exhibit.

In the sliding mode, the friction force along the contact surface between the subunits dominates the behavior (Fig. 3a). This is in the direction orthogonal to the force chain. The subunits are jammed until a slip force is reached due to the applied loads. The slip force can be remotely increased by increasing the applied magnetic field. In addition, the stick-slip behavior is programmable based on the spacing between the pillars inside the composite. The force at which slip occurs can be approximated as $F_{slip} = \mu_{friction} F_{attraction}$. The attraction force between the two cubes, if approximated as two soft-magnetic dipoles, can be expressed as $F_{attraction} = \frac{\mu_0 H^2 (V\chi_{eff})^2}{4\pi d^4}$, where $\mu_0$ is the permeability of free space, $d$ is the distance between the two dipoles, and $V$ is the total magnetic volume. Note that if everything that depends on size in this equation (namely, $V$ and $d$) is scaled, the attraction force is proportional to the square of the distance, demonstrating that jamming forces scale similarly to other surface-based effects, such as surface tension and electrostatic forces. Considering scaling laws, at smaller scales, jamming forces will overcome volumetric forces, such as gravity. This model approximates how the slip force can be tuned by the applied magnetic field, irrespective of other parameters, as long as the material does not reach saturation. After the slip point, the subunits slide with respect to each other and dissipate energy through friction, resulting in a plastic deformation behavior. During sliding, the pillars within the neighboring composites start misaligning and re-aligning. The force range during this sliding is expected to oscillate between two values, with the maximum force occurring when the neighboring patterns are perfectly aligned and the minimum when the misalignment is maximum. The oscillation's wavelength should be equal to the pillar spacing, which was confirmed experimentally (Fig. 3b–d). Moreover, if the spacing between the pillars is increased (for example, to increase the overall magnetization possible given a specific magnetic field as detailed in the previous section), the force could drop to zero during these oscillations (Fig. 3d). This programmable oscillatory behavior could be desirable (e.g. for acoustic applications that leverage vibrations, robotic position-control applications that benefit from the discretized deformations) or detrimental (e.g. for force-control applications, which wouldn't be able to tolerate the drops in the force). For the latter case, the stick-slip behavior can be smoothed and the drops in the force can be avoided by creating subunits with random tight patterns of magnetic materials, as shown in Fig. 3e. To analyze different design parameters that can be modified to methodically smooth the force-displacement behavior, additional simulations were conducted, and the results can be found in Supplementary Fig. 6.

In the pivoting mode, the attractive force between the subunits in the direction orthogonal to the contact surface dominates the behavior. This is along the axis of the force chain. The subunits are jammed until a breakaway moment is reached, which can be calculated by performing a moment balance about the pivot point, which will result in the relationship $M_{breakaway} = CF_{attraction}$, where $C$ is determined by structure geometry. The same attractive force equation from above can be utilized here, which shows that the breakaway force is tunable by remotely changing the magnitude of the applied magnetic field. After the breakaway, the force decreases as the distance between neighboring subunits increases. This can be again modeled with a dipole approximation (red-dashed line in Fig. 3f). For very small deformations, this is enough to describe the mechanical behavior. However, with larger deformations, as the subunits' easy axes significantly deviate from the direction of the applied magnetic field, the external magnetic torque applied to these subunits starts to influence the behavior. The applied torque increases as the pivot angle ($\theta$) increases and can be described by: $\tau = \mu_0 V \mathbf{M} \times \mathbf{H}$ (green-dashed line in Fig. 3f)[26]. When these two magnetic forces are considered, a fall and a rise in the force after the breakaway is predicted by the model, which was confirmed by experimental results (Fig. 3g). The deformation is elastic since at larger displacements the external magnetic field applies a torque to realign the subunits, and at smaller displacements, the attractive force between the subunits assists reassembly.

The separating mode is a simpler version of the pivoting mode, where the attractive force between the subunits also dominates the behavior. The subunits are jammed until a separation force is reached, which occurs when the force separating the subunits is equal to the attraction force between them: $F_{separation} = F_{attraction}$. After breakaway, the force decreases exponentially as the distance between neighboring subunits increases.

In a jamming structure under an applied load, one or multiple deformation modes can occur along any of the contact surfaces between subunits, and the models proposed can help predict the behavior. First, the internal loads along each of the contact surfaces have to be determined based on continuum mechanics, assuming that the entire jamming structure is one solid material. Figure 3h shows the internal loads along a single contact surface. Once the local internal tensile force ($F_x$), the shear forces and torques ($V_x$, $V_y$, $T_x$), and the bending moments ($M_x$, $M_y$) are determined as a function of the applied load, they can be compared to the separation force, the slip force and the breakaway moment, which were defined, respectively. If the internal tensile force is greater than the separation force, the neighboring subunits will separate. If the internal shear force is greater than the slip force, the neighboring subunits will slide. And lastly, if the internal bending moment is greater than the breakaway moment, the neighboring subunits will pivot.

For example, if we take a simple one-dimensional magnetic-jamming structure of length $L$ in which all the subunits are identical, all the inter-subunit forces (separation, slip, breakaway) will be identical throughout the beam, allowing the analysis to focus on the points along the beam where the internal loads are at a maximum. The jamming-based beam is loaded with a point load ($P$) at the center. To demonstrate different internal load conditions, two different boundary conditions were tested. First, a simple three-point bending case was taken. Here, the maximum internal shear force in the beam had a magnitude of $P/2$ and the maximum bending moment in the beam was at the center of the beam with a magnitude of $PL/4$. The slip force was $F_{slip} = \mu_{friction} F_{attraction}$, and the breakaway moment was $M_{breakaway} = (a/2) F_{attraction}$, where $a$ was the height and width of each subunit (8 mm). With an estimated value of 0.5 for the friction coefficient, the breakaway force was predicted to be reached sooner than the slip force, such that the dominant deformation mode would be pivoting and would originate at the center of the beam. This was confirmed with experiments, as shown in Fig. 3i. The beam was also tested in a double-clamped configuration. In this case, we expected the maximum shear in the beam to again be $P/2$, but the maximum moment in the beam was $PL/8$. Mirroring the calculations above, we

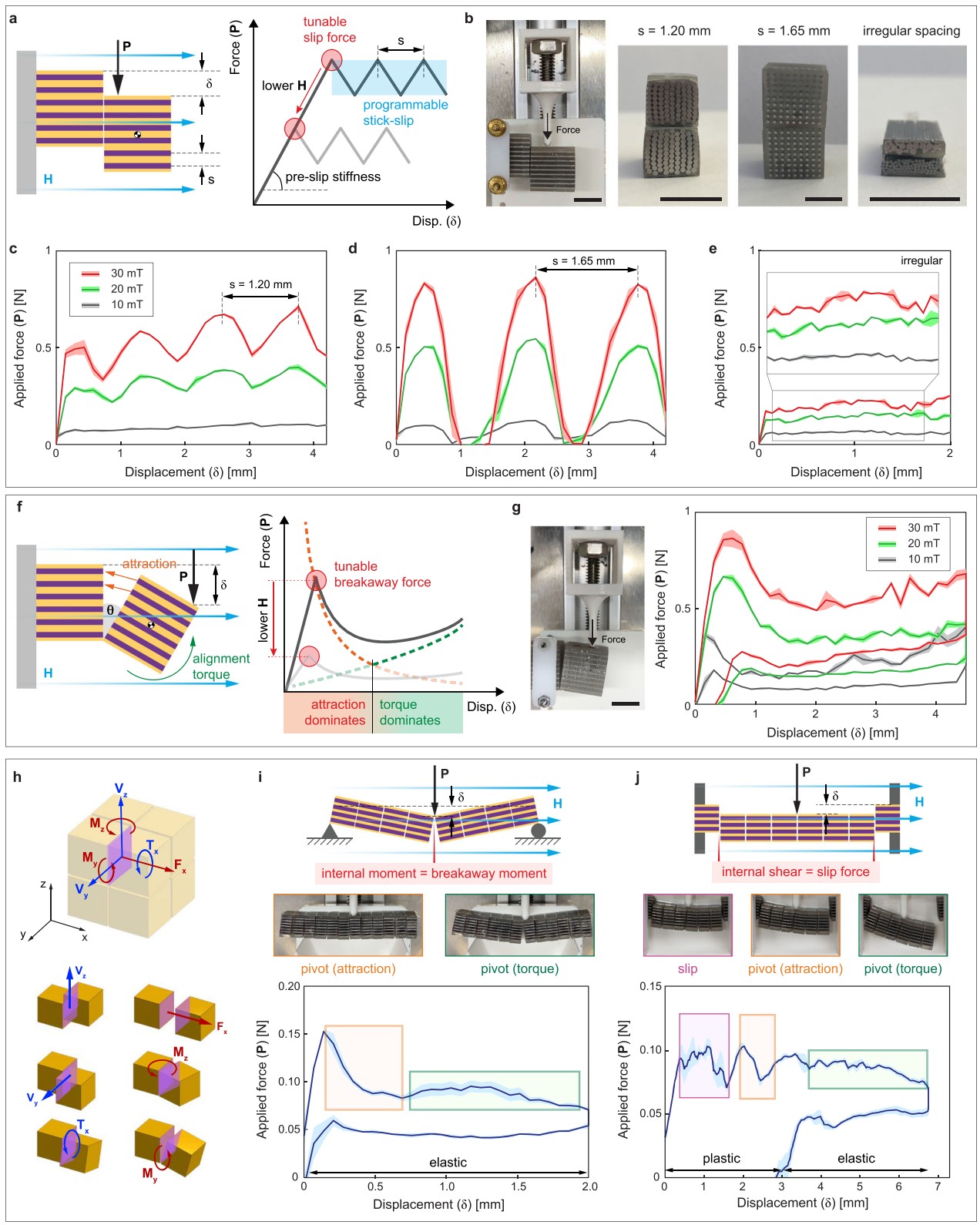

then predicted the dominant deformation mode to be sliding. This was also confirmed with experiments, as shown in Fig. 3j. As the structure slid out, the beam experienced a slightly stronger pull force from one side due to imperfections in the structure and the load applied, and started to exhibit a cantilever-like configuration. The change in boundary conditions increased the maximum shear to $P$ and the maximum bending moment to $PL/2$. This switched the deformation

mode to pivoting with the origin point on the side that was still connected.

As demonstrated in these examples, the force-deformation behavior of magnetic-jamming structures can be predicted by combining the magneto-mechanical models introduced here with continuum mechanics models. While the examples only show the 1-D cases with identical subunits for simplicity, they can be extended to 2-D and

**Fig. 3 | The mechanics of a jamming structure are examined by the main interaction modes between two subunits. a** The sliding mode demonstrates a tunable slip force with a programmable stick-slip behavior. **b** Different pillar densities and distributions were tested (Scale bars: 1 cm). **c, d** Force-displacement curves show that the slip force is tuned by changing the external magnetic field (H) and that the stick-slip behavior is pre-programmed with the distance between the pillars. **e** The stick-slip behavior can be smoothed with irregular pillar spacing. **f** The pivoting mode demonstrates a tunable breakaway force. The attraction between the subunits and the torque applied on the individual subunits by the external field have a combined influence on the force-displacement behavior. **g** The force-displacement curve shows the tunable breakaway force as well as the external

torque effect (Scale bar: 1 cm). **h** The interaction modes are activated within a structure based on the internal loads induced along contact surfaces. Specifically, internal shear forces and torques are working against the slip force, internal bending moments are working against the breakaway force, and internal tensile forces are working against the separation force. **i** A one-dimensional jamming structure is loaded in a three-point bending setup. Given the internal load distribution, it is predicted that the pivoting mode will activate at the center of the beam where there is maximum bending moment. **j** A one-dimensional structure is double-clamped and loaded with a point load at the center. Given the internal shear forces and bending moments, the structure is expected to slip first. See Supplementary Movies 3 and 4 (all shaded regions in the plots show the standard error).

3-D structures with locally varying subunit designs. To handle such complexity, the models proposed could either be integrated into finite element analysis (FEA)-based simulations as constitutive models or integrated into multi-rigid-body models as additional inter-body forces. This enables the creation of complete forward models for more complex structures, which then leads to inverse models, enabling programmable force-deformation regimes.

The ability to model, predict, and program the mechanical behavior also allows engineers to take advantage of the different force-deflection behaviors each interaction mode demonstrates. While the sliding mode offers plastic deformation, energy dissipation, and shape memory, which are advantageous for applications, such as molding or damping, the pivoting and separating modes offer elastic deformation and shape recovery, which are advantageous for applications, such as force control and sensing.

## Multi-directional jamming

One-dimensional magnetic-jamming behavior can be extended to multiple dimensions as briefly introduced in the first section. A composite can be designed with multiple discrete easy axes, each of which can be selectively magnetized depending on the orientation of the applied magnetic field. This configuration allows real-time adjustment of jamming directionality, determined by the directional options encoded in the composite design. For planar jamming, this could be done by embedding a cross-shaped magnetic body that is equally elongated in the x and y directions, or by creating two orthogonal magnetic pillar patterns within the composite in both the x and y directions. When a magnetic field is applied at a 45-degree angle to the x-axis, the structure equally jams in both dimensions. In contrast, if a field is applied only along x, the structure will only jam in the x-axis, and if the field is only applied along y, the structure will only jam in the y-axis (Fig. 4a). The ability to independently jam a structure along different directions may offer greater flexibility in controlling mechanical responses compared to other actuation methods.

Force chains within jamming structures are formed in the direction of the applied magnetic field, as long as the subunit composites have an easy axis along that direction. When the jamming structure is planar or volumetric, repulsive forces between adjacent subunits should also be considered. Depending on the subunit design, these forces are induced in the direction perpendicular to the field, as can be observed in the Supplementary Movies. For example, if we apply a field in the x direction in a 2-D structure (as in Fig. 4a), the subunits might repel each other in y. These repulsion forces can be leveraged (for example, by geometrically constricting the entire aggregate with an encasing and utilizing the repulsive force chains for stiffness control) or minimized (for example, by concentrating all the pillars in the center of a subunit, increasing the distance between the individual repelling elements between neighboring subunits). The versatile multi-degree-of-freedom behavior was demonstrated with lifting experiments in Fig. 4b. As shown, force chains can be formed along lines (in x, y, or z), along planes (xy, yz, or xz), and along volumes (xyz) in a three-dimensional aggregate. This multi-dimensional tunability can be leveraged to create actively tunable mechanical behaviors for jamming

systems (both for static and dynamic applications), as well as to remotely disassemble and assemble multi-dimensional structures in different sequences.

This directional control can also be leveraged to create structures that have anisotropically tunable stiffness. To demonstrate this, a two-dimensional grid of magnetic subunits was bonded to a strain-limiting layer on one side (Fig. 4c and Supplementary Fig. 5). The strain-limiting layer in the structure constrained the interaction mode between subunits to pivot (as explained in the previous section). When there was no applied field, the structure exhibited low stiffness in both dimensions and deformed under its own weight. When a field was applied in one of the two main axes of the structure, the structure stiffened in that direction. The stiffened direction demonstrated the mechanical behavior outlined in the previous section (Fig. 3f), and the strength of the sheet can be tuned remotely in multiple dimensions. As discussed in the first section, a magnetic field can also be applied at a 45-degree angle between the two main axes of the structure, equally stiffening and jamming in two dimensions. Furthermore, the field can be applied at different angles to stiffen one axis more than the other.

## Interaction with objects and environment

With their capacity to be remotely actuated by magnetic fields along pre-programmed easy axes, magnetic-jamming structures enable versatile interaction possibilities with objects and their environment. These capabilities are especially promising when integrated with rapidly advancing electromagnetic navigation systems, which allow multi-dimensional control within their workspace[27–31]. To demonstrate the core mechanical interaction functions, three proof-of-concept examples are presented: a gripper performing a pick and place task, a dilation beam reversibly anchoring to a soft environment, and a distributed jamming structure interacting with a medium intertwined throughout it to demonstrate complex topological adaptation. The gripper and distributed jamming structure further illustrate how jamming actuation can operate in conjunction with other untethered magnetic actuation modes, such as rotation and translation.

The 1-D magnetic-jamming gripper is created with three subunits, all connected with a single strain-limiting layer. Two subunits make the fingers of the gripper, while one makes the palm. Soft-ferromagnetic pillars are placed along the direction of grasping. The strain-limiting layer provides a pre-tensioning that allows for the gripper to have a default open state, and the overall structure behaves in the pivoting deformation mode. The gripper closes and applies an increasing amount of grasping force as the applied magnetic field increases. Since the gripper has a preferential easy axis, the gripper can easily be rotated by changing the direction of the applied magnetic field, without changing the grasping force. Additionally, the gripper can be moved in space by applying magnetic field gradients, and depending on the magnetic field magnitude, this can be done in the open or closed state of the gripper. These functionalities are demonstrated in a pick and place task in Fig. 5a. The magnetic-jamming gripper can approach and gently grasp a soft object (a small sponge piece), carry it through an L-shaped path, and release the object at a desired

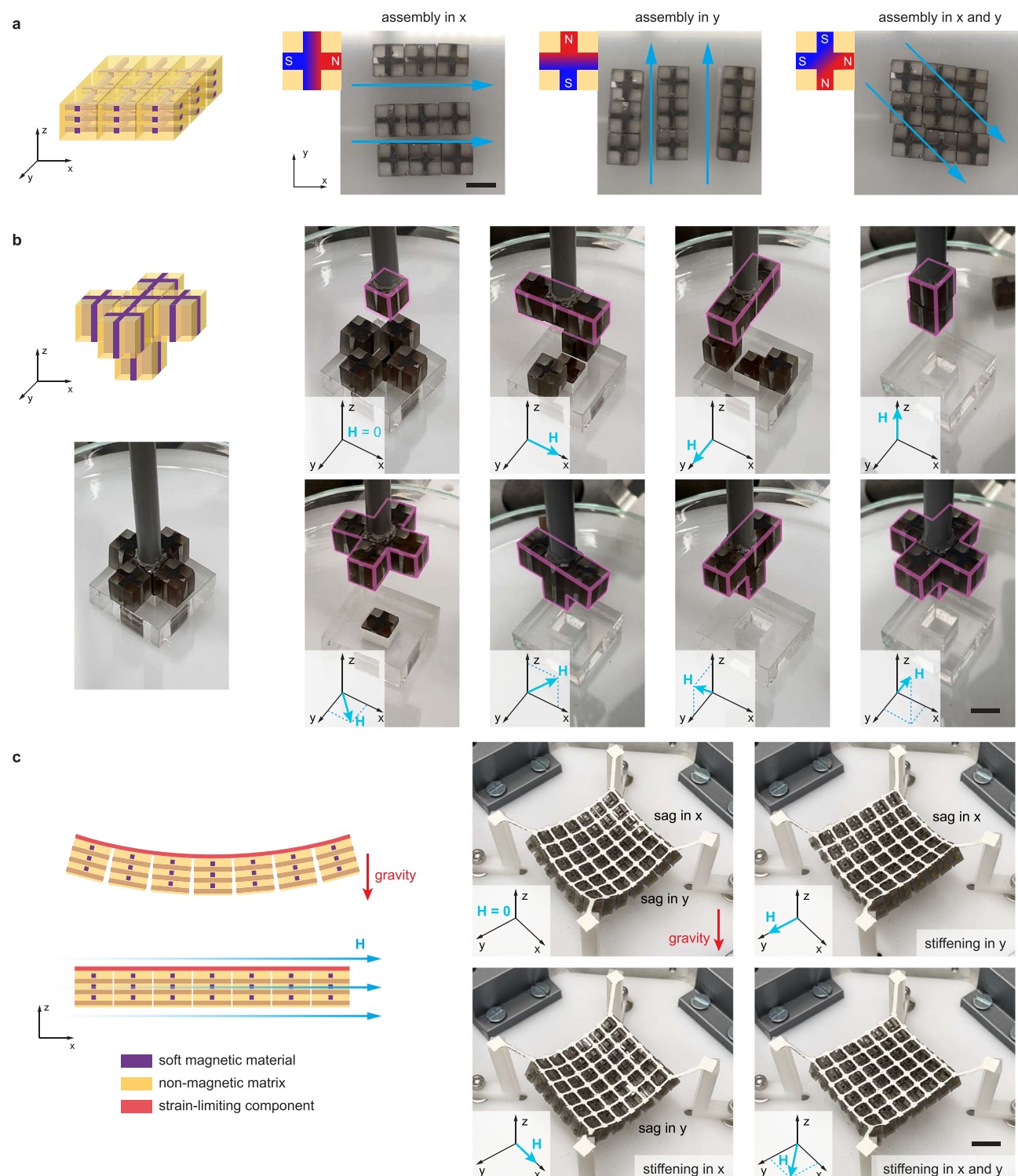

**Fig. 4 | Magnetic composites induce jamming selectively along specific directions within a multi-dimensional structure. a** 3-by-3 two-dimensional structure in the x-y plane can be jammed only in x, only in y, and in both x and y, based on the direction of the applied magnetic field (H). **b** A lifting experiment demonstrates how a three-dimensional structure can be jammed in all possible combinations of the three dimensions (none, x, y, z, xy, xz, yz, xyz). **c** Magnetic composites also enable multi-dimensional stiffness-tunability. See Supplementary Movies 5, 6, and 7 (Scale bars: 1 cm).

destination. The gripper is 3 mm thick, and along the grasping plane, it can either have a dimension of 6 mm by 6 mm when fully closed, or of 3 mm by 15 mm when fully open (Supplementary Movie 8). The gripper is smaller than typical wireless capsule endoscopes, indicating the potential for integrating similar jamming structures into conventional endoscopy and/or colonoscopy procedures to enhance functionality[32].

The dilation beam is made with seven cubic subunits (each $3 \times 3 \times 3$ mm) that are all connected with a strain-limiting layer (Fig. 5b). When unjammed, it conforms to the soft environment around it. When a magnetic field is applied and slowly increased, it gradually straightens along its easy axis and stiffens. It is able to deform the soft environment around it and anchor itself. When the magnetic field is

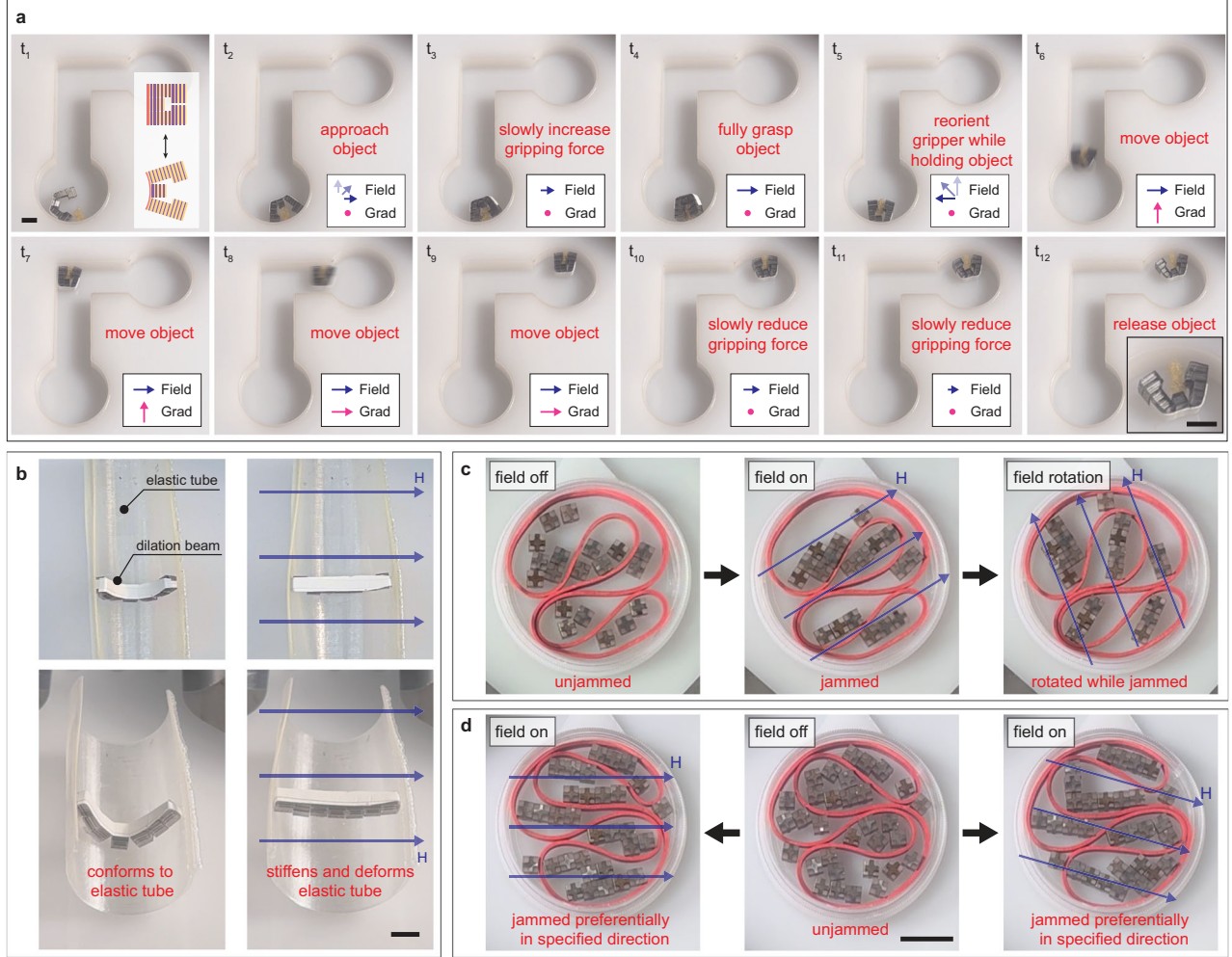

**Fig. 5 | Case studies of magnetic jamming systems in contact with other objects and the environment. a** A magnetic gripper taking advantage of the pivoting deformation mode between magnetic-jamming subunits, where gripping is controlled by magnetic field magnitude, orientation by magnetic field direction, and motion by magnetic field gradients (Scale bar: 5 mm). **b** A dilation beam that can conform to soft medium around it, or can gradually jam and stiffen as it gets into its pre-designed shape to deform medium and anchor itself (Scale bar: 5 mm). **c** A magnetic-jamming structure can be distributed across a medium, achieving distributed jamming, a special feature of magnetically induced jamming. **d** When the packing density of subunits is higher, better multi-directional jamming is observed. See Supplementary Movies 8, 9, and 10 (Scale bar: 10 mm).

removed, it can again conform to the environment (Supplementary Movie 9).

Untethered jamming is advantageous not only because the external actuator (electromagnetic navigation system) can remotely control the jamming behavior, but also because a jamming structure need not be a single cohesive structure. A structure can be divided and distributed around the environment and foreign objects. This behavior is demonstrated by randomly dispersing 2-D jamming subunits within a soft medium—specifically, an elastic band forming relaxed, self-contacting loops. When a magnetic field is applied, the subunits attract each other along the direction of the applied magnetic field, forming jammed sub-structures that can exert forces on the surrounding elastic band. The magnitude and orientation of these contact forces depend on several factors: (a) the magnitude of the magnetic field, (b) the alignment of the units with respect to each other, (c) their orientation with respect to the applied magnetic field, and (d) interactions with the surrounding medium. The location, orientation, and magnitude of these contact forces can be dynamically adjusted by rotating the magnetic field while the system is jammed or by switching the field off in one direction and reactivating it in another (Fig. 5c and Supplementary Movie 10). Additionally, a higher packing density of subunits enhances multi-dimensional jamming by reducing the gaps between them, leading to more effective and multi-directional force transmission (Fig. 5d and Supplementary Movie 10).

## Discussion

A class of magneto-mechanical metamaterials based on jamming is presented, in which the jamming transition can be controlled remotely and directionally. These metamaterials demonstrate versatility and allow for programmability in terms of: active remote control of stiffness and yield, 1-D to 3-D control possibilities based on design, isotropic or anisotropic mechanical behaviors, elastic and/or plastic deformations, and stick-slip behavior. While this paper mainly focuses on ordered systems, the experimental findings, analytical models, and simulation frameworks can be extended to stochastic aggregates as briefly demonstrated in Fig. 5c and the Supplementary Movie 10.

The ability to actuate the structures in an untethered manner, without requiring tubing or wiring, enables scalable jamming structures that can operate in hard-to-reach constricted areas, such as inside the human body and inside structural infrastructure, such as pipes and tubes. For example, variations of the dilation beam shown in Fig. 5b can be used as an anchor for operating tools, enabling local stability at a desired location. The beam can be remotely stiffened or softened along one or more directions, all based on the task at hand.

Multi-dimensional control possibilities also enable structures that can be softened in the direction where force control is required and stiffened along the dimension that requires position control. This type of multi-degree-of-freedom control can also be integrated into applications of magnetorheological-fluid-based devices, such as valves[33].

The proposed composite-based jamming structures also allow local programmability of inter-subunit forces through design at the subunit level. While the present study has focused on homogeneous jamming structures with identical subunits, the work can be extended to heterogeneous jamming structures that have subunits with different directionalities and magnetization properties. This enables spatially programmable mechanical properties with the ability to guide force chains along specific directions and specific elements. Conventional jamming structures (such as those actuated by vacuum) do not have this degree of programmability and are therefore functionally more limited. By utilizing micro-electro-mechanical system (MEMS) fabrication technologies that can create high precision micro-scale magnetic arrays[34–37], the proposed jamming technology can enable the creation of micromanipulation platforms with locally varying stiffness or yield at the micrometer scale for applications, such as active media to study the influence of mechanics on cell growth[38,39].

The proposed magnetic-jamming structures are highly modular and reconfigurable, unlike prior jamming structures utilized in robotic applications. Specifically, subunits can be added and removed quite easily from an aggregate, without an additional step. For example, for vacuum-actuated jamming, the airtight casing needs to be opened and closed after new subunits are added or removed. For electrostatic jamming, each new subunit needs new wiring. In magnetically induced jamming, a new subunit could simply be added to the aggregate without any changes made to the overall structure. The proposed system also contributes to existing magnetically actuated tunable structures that are not based on jamming, as they primarily consist of continuous structures that cannot assemble or disassemble[40–45]. It also contributes to existing modular magnetic systems, which utilize hard-magnetic materials and cannot be easily disassembled, by utilizing soft-ferromagnetic materials, which can easily be demagnetized[46–48]. The reconfigurability makes magnetic jamming especially suitable for systems that require structures that can be reconfigured in situ while maintaining their tunability.

This study focuses on inducing specific jamming behaviors based on applied magnetic fields, where the subunits have fixed magnetic properties. Prior work has demonstrated how the bulk magnetic response of a composite can be reprogrammed, such as by thermally softening the surrounding non-magnetic matrix to enable the realignment of the magnetic components[49,50]. The magnetic response of a composite has also been modified by utilizing jamming, through swelling a surrounding porous polymer around magnetic particles within the pores[51]. Future work could combine these works to achieve dynamic magnetic properties of the subunits, which would allow for further actuation versatility and introduce additional degrees of freedom for control.

The magneto-mechanical behaviors of magnetic-jamming structures could be particularly suitable for applications in: biomedical devices, microassembly and micromanipulation platforms, infrastructure monitoring devices, shock absorption systems, or other applications where sensing and actuating must be done remotely. Since the primary forces considered are magnetic, the structures can also work in low or zero-gravity environments, allowing the resulting technologies to be translated to space. The magneto-mechanical analytical models, validated through experimental results and numerical simulations, provide a basis for realizing versatile combinations of magnetic, structural, and mechanical functional properties.

While this work presents and validates key aspects of untethered, directionally programmable magnetic jamming, several important limitations should be acknowledged. First, although we demonstrate that jamming and unjamming are reversible in controlled assemblies, achieving reliable reassembly of fully untethered subunits—especially in stochastic or dynamic environments—remains a significant challenge. Assembly fidelity and positional consistency may require additional mechanical design strategies, such as linking mechanisms or environmental constraints. Alternatively, a more statistical, non-deterministic approach may be viable, where only general assembly trends are targeted in multi-subunit systems. Second, while we outline approaches for downscaling and show that our current designs operate within biologically relevant dimensions, reliably and repeatably fabricating increasingly complex composite patterns at smaller scales remains nontrivial, particularly when multi-material integration is needed. Material choices will also need to account for biocompatibility in the context of biomedical applications. In addition, while we demonstrate the feasibility of combining jamming with translational and rotational motion using magnetic fields, precise multi-degree-of-freedom control remains a complex challenge. Our demonstrations offer qualitative validation using open-loop control; future work will be needed to implement and evaluate robust, quantitative, and closed-loop control in dynamically changing environments. Addressing these limitations will be important for enabling real-world applications, especially in biomedical or other micro-scale domains where reproducibility and precision are critical.

## Methods

### Materials and manufacturing

All subunits for the jamming structures were fabricated with 304 stainless steel as the magnetic material and fixed in a clear epoxy resin matrix (Ultra Clear Pro, Epodex). Silicone molds (Dragon Skin, Smooth-On Inc.) were used to fabricate these composites. The desired magnetic material pattern was placed inside the molds, and a two-part liquid epoxy resin was poured in to fill the spaces left by the magnetic pattern. The resulting composite was removed from the mold after the resin was cured and the pattern was fixed in place. The magnetic patterns for the 1-D jamming subunits were fabricated using off-the-shelf 304 stainless-steel wires (McMaster Carr) that were placed into laser-cut acrylic scaffolds for precise positioning. For the 2-D jamming subunits, carbon steel sheets were cut using a waterjet (ProtoMax, OMAX Corporation) to create the desired individual layers of patterns. These layers were then stacked inside the mold using spacers before the resin was poured. The silicone molds for the 1-D and 2-D subunits enabled the fabrication of large continuous composites. These were then cut into smaller pieces using a waterjet. For the 3-D jamming subunits, the composites were cast individually, since the pattern could be created from a continuous piece of magnetic material. These parts were created using a waterjet on carbon steel plates. Each individual magnetic piece was placed inside the mold. The samples were directly used after being released from the mold (Supplementary Fig. 2).

### Magnetic characterization

A vibrating sample magnetometer (MicroSense EZ7 VSM, MicroSense LLC) was used to characterize the magnetic properties of the samples. Each sample was created to fit in an 8 mm disk and was magnetized along the in-plane direction. For each sample, the full hysteresis loops were obtained, magnetizing it from −10000 to 10000 Oersted.

### Actuation with a magnetic field

A 5-DOF electromagnetic system with eight magnetic coils was used (OctoMag, Magnebotix AG). In all experiments, a constant magnetic field of a given magnitude was generated in the direction along the easy axes of the samples, while maintaining zero magnetic gradient. A zero (or minimum) magnetic gradient allows for experiments to only study the magnetic interaction forces between the subunits, without the influence of forces applied by the external magnetic field. Similarly,

magnetic fields were always aligned with the easy axis of the subunits, ascertaining that the magnetic torque applied to the individual subunits was also zero.

## Mechanical characterization

To measure the forces between the individual subunits, a 3-axis force/torque sensor (Nano17, ATI Industrial Automation) was used. A sample holder was created such that one subunit was fixed with respect to the magnetic field generator (OctoMag, Magnebotix AG)[28]. The other subunit was fixed to the same plate but connected with two linear positioning stages to assist with alignment and gap control. The sample holder was also created such that the force sensor was as far as possible from the workspace so that it was minimally influenced by the applied magnetic field (Supplementary Fig. 3).

A custom mechanical characterization device (Supplementary Fig. 4) was created to collect the force-deflection data. The experimental setup was fabricated such that the sample would be fixed at the center of the workspace of the magnetic field generator. An aluminum 2 N range load cell with 0.1% accuracy (KD24S, ME-Messsysteme) was used to collect the force data. The load cell was attached to a motorized linear stage with a 10 cm stroke (drylin® linear module: SLNV-27 DS6.35×2.54, igus® Schweiz GmbH). A 3D-printed connector was screwed onto the other end of the load cell to push onto the sample. All devices were controlled using the Robot Operating System (ROS). Since the force data from the sensor and the position data from the motor driver were at different frequencies, the timestamps of each data point were used to create a linear interpolation, so that the output data from each experiment directly had a force and a displacement data point for each timestamp. The resulting force-displacement curves presented are based on these interpolated points.

## Analytical modeling

For the analysis of the magnetization behavior, an ellipsoid approximation was used for the individual pillars to calculate the demagnetizing factors[52]. Specifically, given an aspect ratio of an individual pillar of $R = l/d$, where $l$ is the length of the pillar and $d$ is the diameter, the demagnetizing factor ($n_l$) along the length is determined by the equation: $n_l = \frac{1}{R^2-1}\left(\frac{R}{2\sqrt{R^2-1}}\ln\left(\frac{R+\sqrt{R^2-1}}{R-\sqrt{R^2-1}}\right) - 1\right)$. The component of the susceptibility along this dimension then becomes $\chi_{pillar} = \chi_{material}/n_l$.

When analyzing the magnetic interaction between two pillars, each pillar was approximated as a dipole point at the centroid of the pillar. The magnetic field induced on each pillar by the neighboring pillar was approximated to be homogeneous throughout the entire pillar and was modeled to have an equal demagnetizing effect on each other.

For the mechanical analyses, all of the subunits were modeled as rigid bodies. While one subunit was kept fixed, a force and moment balance on the subunit was performed. Magnetically, each subunit was considered as a single dipole located at its centroid, with the effective susceptibility numerically calculated utilizing simulations. The intersubunit forces, as well as the externally applied magnetic torques, were calculated.

## Numerical simulations

Magnetization simulations were run in COMSOL 6.2 using the magnetic fields module. 3D models were created for the composites. The cylindrical pillars were simulated to have a relative permeability of 200, and the surrounding matrix was specified to have a relative permeability of 1. A large cylindrical body of air containing the composite was also included in the model. To simulate the externally applied magnetic field, a magnetic flux was assigned to the two circular ends of the cylinder of air. This direction aligned with the intended easy axis of the composite, the major axes of the pillars.

To extract the magnetization of the composite, a volume average of the magnetization was taken, specifically looking at the component of the magnetization along the direction in which the magnetic field was applied. The effective susceptibility was then calculated as the ratio of the resulting magnetization ($M_z$), over the magnitude applied magnetic field ($H_z$).

## Data availability

All data supporting the findings of this study are available within the article's supplementary files. The source data for all the data used to make the figures is provided in a single Excel file. Any additional requests for information can be directed to and will be fulfilled by the corresponding author(s). Source data are provided with this paper.

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

## Acknowledgements

We thank Alexandre Mesot, Claas Ehmke, Erdem Siringil, Felix Grüninger, Guido Nafz, and Bernard Javot for their assistance regarding sample fab-rication and experimental setup construction, and Elizabeth Zuurmond for proofreading this article; and members of the Harvard Biorobotics Lab and Andra Constantinescu for valuable discussions. This work was primarily supported by an ETH Zurich Postdoctoral Fellowship received by B.A. Additionally, M.K. acknowledges partial financial support by the Swiss National Science Foundation under Project No. 200021L_197017.

## Author contributions

B.A. conceived the project, performed the analytical modeling, the COMSOL simulations, the magnetization characterizations, and the mechanical behavior experiments. B.A., M.B., and G.S. fabricated the samples. B.A., M.K., M.B., and G.S. designed the experiments. M.B. conducted the funnel and stiffness-tunability experiments. M.B. and B.A. conducted the lifting experiments. G.S. and B.A. conducted the inter-action force experiments. G.F. and F.H. built and validated the device for mechanical characterization. B.A., M.K., and M.B. prepared the figures and videos. B.A. wrote the manuscript, all authors interpreted the results, and M.K., M.B., S.G., X.C., S.P., and B.J.N. reviewed and edited the manuscript.

## Competing interests
The authors declare no competing interests.
