## [Transparent Peer Review file · Nature Communications]

Jamming with Magnetic Composites

Corresponding Author: Dr Buse Aktaş

Version 0:

Reviewer comments:

Reviewer #1

(Remarks to the Author)

This work introduces an untethered jamming mechanism based on soft ferromagnetic composites. The study begins with a detailed demonstration of the fabrication strategy and actuation mechanism, highlighting the jamming/unjamming phenomenon induced by remote control of magnetic field. Next, both analytical and numerical studies are presented to establish design rules for predicting mechanical properties of the jamming structures, in which three distinct modes may occur under applied external loadings. Finally, experimental results are provided to demonstrate multi-dimensional jamming in 1-D, 2-D and 3-D structures, showing the potential for selectively jamming and shape morphing. While these creations are promising, additional experimental validations are needed to further support the claimed innovation of the untethered jamming structure.

COMMENTS:

1. This article emphasizes the innovation of the untethered jamming structure, particularly the miniaturization and shape morphing capabilities, which represent significant advances over traditional vacuum-based jamming structures (as discussed on Page 2). However, the unit cell size of the proposed untethered jamming structure (~ 1 cm) is considerably larger than that of existing vacuum-based jamming grippers (e.g., Brown E et al. PNAS 107:18809-14(2010)) which features much smaller unit cells. This discrepancy raises concern about the practicality and scalability of the proposed mechanism, particularly for biomedical applications like surgical robotics and micromanipulation. To substantiate the novelty of this untethered jamming approach, the authors should present experimental results demonstrating magnetic jamming structures with smaller unit cells that could be competitive with the vacuum-based jamming systems. Furthermore, while the shape morphing results in Figure 4 are interesting, they appear relatively simple compared to the more complex and random shape-morphing/locking behaviors achieved by tethered jamming grippers. The authors should provide additional examples of more intricate and diverse shape-morphing/locking results.
2. On Page 14, the authors analyze the oscillatory behavior observed in sliding mode. This oscillatory problem could be addressed by incorporating subunits with randomly distributed magnetic composites. The authors may provide optimized results demonstrating the most smoothed force-displacement curve the structure can achieve.
3. Some error bars are missing in figures, e.g., Figure 3i and j. The authors should include error bars in all relevant figures to provide comprehensive results.

Reviewer #2

(Remarks to the Author)

This article titled "Jamming with magnetic composites" studies an externally controllable and tunable jamming phenomenon of centimeter-scale units. The jamming, or shape-locking, could be achieved along one dimension, in a plane, or in a volume. One difference between the proposed jamming strategy with other existing ones is that it is untethered and wirelessly triggered using a magnetic field. The working principle of this jamming strategy is based on the magnetic properties of soft ferromagnetic particles with low remanence and low coercivity. When an external magnetic field is applied, these magnetic particles are magnetized and they generate attractive forces between each subunit, causing jamming. When the magnetic field is removed, these particles do not retain much magnetization and thus the attractive force disappears, leading to unjamming. This article also characterized the force-deformation behaviors of the jamming structures using analytical models, numerical studies, and experimental validations. The article mentioned that the targeted applications of

the proposed jamming mechanism include surgical robotics, micromanipulation, and structural monitoring. The reviewer has the following questions and comments.

1. The main novelty of this work is an untethered jamming mechanism. This article compares the proposed jamming mechanisms with some existing ones, especially the one applies a vacuum to the aggregate with an airtight envelope and the one utilizes a voltage. Then this article claims the key advantage of the proposed one is that it is untethered and easy to scale and reconfigure. However, there lacks a comparison between the proposed jamming mechanism with existing untethered ones. For example, there are previous studies on magnetically induced jamming. Here are some references:

- a) Khalil ur Rehman et al., Magnetic microactuators based on particle jamming, ACS Materials Letters 2024.
- b) T Leps et al., A low-power, jamming, magnetorheological valve using electropermanent magnets suitable for distributed control in soft robots. 2020.
- c) Xianhu Liu et al., Magnetic field-driven particle assembly and jamming for bistable memory and response plasticity, Science Advances, 2022.
- d) Xianhu Liu, Magnetic field-induced particle assembly and jamming, Doctoral thesis, Aalto University, Finland. 2024.

2. Another key novelty in this article is the ability to selectively jam structures along specific directions. Is it the case that one specific kind of subunits can only be jammed along a predefined direction and cannot be altered after fabrication? It is hinted in the article that the proposed jamming mechanism can overcome the limitation of only having one “easy axis”. But based on the presented results, it seems like that one type of subunits can only be jammed along one direction. If another direction is desired, a different type of subunits need to be deployed. It is obvious the jamming direction can be programmed by varying the design parameters. But once the subunits have been fabricated, i.e., all the parameters are fixed, can this jamming direction still be altered on-the-fly?

3. This article states that an important motivation and application of this jamming mechanism is to assist robots in hard-to-reach space inside human body. But the prototypes presented in the current manuscript are at the scale of tens of centimeters. They are way too large to enter human body. And there aren't any straightforward methods or clear demonstrations that these prototypes could be downscaled by one or two orders of magnitude.

4. An important limitation of the proposed jamming mechanism is that it requires continuous power consumption to maintain the applied external magnetic field. How does the proposed strategy compare with existing jamming mechanisms from the energy consumption perspective? One of the mentioned applications is untethered robots in minimally invasive healthcare, many of which rely on magnetic actuation. How does this jamming mechanism work alongside those magnetically controlled robots? Will they be compatible in the same workspace? Or maybe this jamming mechanism can only work with robots that use a non-magnetic working principle? It is an important point to clarify.

5. This article claims that the jamming is reversible (page 6 line 101). But all the demonstrations, especially the ones shown in the supplementary videos, do not contain reversible jamming and unjamming. For example, in supplementary video 1, the 1-D, 2-D, and 3-D tests start with an already jammed state and become unjammed when the magnetic field is removed. Will they be jammed again if the magnetic field is applied afterwards? It looks like the jamming state can only be achieved when all the subunits are well aligned to start with. In other words, the subunits may need to be manually placed into a certain pattern before applying a magnetic field to induce jamming. If that is the case, then this jamming-unjamming transition is not reversible and this point should be elaborated in the manuscript.

6. Page 5 - Line 91-94: “The effective susceptibility of the subunits (χ_{eff}) can be determined experimentally or numerically by $H = M \chi_{\text{eff}}$, where M is the resulting magnetization of the subunit, and H is the applied external field. [14]” This equation is wrong. It is unclear if this error is just a simple typo, or this wrong equation is used in the development of the analytical model and numerical calculations. Since this mistake is repeated in Methods - Numerical simulations (page 29 line 637), the reviewer is worried that this mistake is probably not just a typo and could have caused a systemic miscalculation in the analytic and numerical models.

7. What are the soft-ferromagnetic materials exactly? It is not mentioned in the Methods section or SI.

8. It lacks obvious applications of the proposed jamming mechanism. Lots of potential applications are discussed in the section of Discussion. But none of these mentioned applications are demonstrated or can be directly inferred intuitively from the demonstrations presented in the current manuscript.

Reviewer #3

(Remarks to the Author)

See Comments for Author in the attached PDF.

Version 1:

Reviewer comments:

Reviewer #1

(Remarks to the Author)

I have reviewed the revised manuscript entitled "Jamming with Magnetic Composites" and the authors' responses to the initial comments. I appreciate the thorough efforts in addressing all the concerns raised during my first review. The revisions, including the innovation concerns compared with the vacuum-based jamming techniques and oscillations phenomenon, have significantly improved the clarity and quality of the manuscript. The key points have been adequately addressed, and this work is now ready for publication.

Therefore, I recommend that the manuscript be accepted for publication.

Reviewer #2

(Remarks to the Author)

Please refer to the attached PDF file.

Reviewer #3

(Remarks to the Author)

The authors have made sufficient modifications in response to the reviewers' comments. In particular, they have adequately addressed my concerns in the revised manuscript. I have no further comments to improve this work.

Response to Reviewers

Reviewer #1 (Remarks to the Author):

This work introduces an untethered jamming mechanism based on soft ferromagnetic composites. The study begins with a detailed demonstration of the fabrication strategy and actuation mechanism, highlighting the jamming/unjamming phenomenon induced by remote control of magnetic field. Next, both analytical and numerical studies are presented to establish design rules for predicting mechanical properties of the jamming structures, in which three distinct modes may occur under applied external loadings. Finally, experimental results are provided to demonstrate multi-dimensional jamming in 1-D, 2-D and 3-D structures, showing the potential for selectively jamming and shape morphing. While these creations are promising, additional experimental validations are needed to further support the claimed innovation of the untethered jamming structure.

1.0 We appreciate the reviewer finds our results promising, and thank them for their valuable suggestions to support the innovation. However, it seems like they might have overlooked several of the main points presented in the manuscript. We have adjusted wording and added new results (which include additional experiments) to emphasize the unique features enabled by our innovation more explicitly. One main clarification we would like to make is that we do not mention “shape-morphing” in the paper. We do not study or program an active shape-change. The contribution of this work is focused on the jamming transition, which results in tunable isotropic/anisotropic mechanical properties (stiffness, damping, yield/force-threshold, plastic/elastic deformations).

COMMENTS:

1. This article emphasizes the innovation of the untethered jamming structure, particularly the miniaturization and shape morphing capabilities, which represent significant advances over traditional vacuum-based jamming structures (as discussed on Page 2). However, the unit cell size of the proposed untethered jamming structure (~ 1 cm) is considerably larger than that of existing vacuum-based jamming grippers (e.g., Brown E et al. PNAS 107:18809-14(2010)) which features much smaller unit cells. This discrepancy raises concern about the practicality and scalability of the proposed mechanism, particularly for biomedical applications like surgical robotics and micromanipulation. To substantiate the novelty of this untethered jamming approach, the authors should present experimental results demonstrating magnetic jamming structures with smaller unit cells that could be competitive with the vacuum-based jamming systems. Furthermore, while the shape morphing results in Figure 4 are interesting, they appear relatively simple compared to the more complex and random shape-morphing/locking behaviors achieved by tethered jamming grippers. The authors should provide additional examples of more intricate and diverse shape-morphing/locking results.

1.1 We thank the reviewer for this comment concerning comparisons with vacuum-based jamming grippers in terms of size, and the mechanical behavior we have demonstrated in Figure 4 (not shape morphing). While the vacuum-based jamming gripper the reviewer mentions is an excellent example of jamming being utilized for a robotics application, there are a number of factors which make it a limiting comparison. Below, we will highlight points brought up by the reviewer in this comment regarding (a) unit cell size (b) scalability and (c) mechanical versatility.

- a. Jamming structures do not generally need to have small unit cells. One of the great advantages of jamming structures for robotics applications is that they can be designed according to the performance requirements of the desired task. The dimensions of the subunits can be varied to program specific tunable mechanical behaviors, and more and smaller subunits per structure does not always mean better performance. These tradeoffs are different based on the actuation method utilized, and magnetic jamming has its own specific physical constraints. For example, in the case of the simple beam-bending we show in Figure 3i, the deformation is localized to one point due to the stress distributions in the beam. Having smaller but more pieces in this case would not change the resulting deformation, and it would actually diminish the overall jamming force achievable by the subunits, since they end up having less magnetic volume per-subunit. This is quite a different physical phenomenon

compared to vacuum-based jamming, in which the actuation forces are generated externally, but not through the individual subunits themselves.

However, we also understand the reviewer's concern about the practicality of the proposed mechanism. To address this, we have added more explicit demonstrations illustrating specific functionalities for applications including but not limited to medical robotics. In our new section "Interactions with Object and Environment", we have added:

- A magnetic jamming-based gripper that can apply a range of pinching/grasping forces based on the applied magnetic field, and can be moved while jammed. (See Fig. 5a and Supplementary Video SV8)
- A dilation beam that can be deployed while flexible, and can gradually anchor itself and apply force to soft environments with the application of increasing magnetic fields. (See Fig. 5b and Supplementary Video SV9)

In the context of biomedical applications, procedures such as endoscopy or colonoscopy would be a good fit for these new demonstrations in terms of the size of prototypes built. Endoscopes and colonoscopes can have diameters up to 12 mm and, have working channel diameters up to 4 millimeters, which means they can easily deliver tools up to this size. Additionally, wireless capsule endoscopes typically have a size of 11 mm × 26 mm, as described in a review article published in Nature Communications (1). The types of jamming structures proposed in this manuscript could either be integrated into these capsule endoscopes or be delivered with endoscopes or colonoscopes. In light of these applications, we have chosen to create structures with 3-4 mm sized subunits, which are also in a similar size-range to a number of granular jamming studies that also incorporate non-standard subunit designs (2, 3). This size also allows for the easier retrieval of the subunits, as they aren't so small that it becomes challenging to visualize with common medical imaging modalities. We would like to note that these structures are smaller than other magnetically actuated reconfigurable surgical tools in literature such as (4).

- b. We also want to clarify the scalability that magnetic jamming offers. Since it allows for remote actuation, structures can be fully untethered (no wires or tubes attached) and do not require on-board actuation mechanisms (an on-board battery or pump). This essentially allows for effective remote control of jamming structures in areas that are not easily accessible (e.g. lumens inside the human body), as they do not need centimeters of pneumatic tubing, electrical wiring, or an on-board power-source and actuator. If we think about scaling laws, we have an additional advantage. Remembering the dipole approximation from the paper (pg 13, line 255), and taking each component to scale with size (magnetic volume, dipole-dipole distance) taking "a" as a dimensional unit.

$$F_{\text{attraction}} = \frac{\mu_0 H^2 (V \chi_{\text{eff}})^2}{4\pi d^4}, V \propto a^3 \text{ and } d \propto a$$

We see that $F_{\text{attraction}} \propto a^2$.

The magnetic jamming forces will scale similarly to other surface-based effects such as electrostatic forces and surface tension. At smaller scales these forces will more easily overcome volumetric forces, such as gravity. Additionally, since power scales with volume, it becomes increasingly challenging to have on-board actuation at smaller scales. Therefore, the ability to achieve untethered magnetic jamming allows us to overcome these physical limitations. (Some additional text has been added in line with this point - see lines 302-205)

- c. As explained above (in response 1.0), the present study does not focus on the shape-morphing aspect of the structures, but on achieving intricate and diverse changes in mechanical properties such as isotropic/anisotropic stiffness and yield. In Figure 4, we do not demonstrate shape-morphing, but we demonstrate the directional control of jamming structures which has not been achieved before. The structures have a single preferred geometry in the jammed state. The novel versatility we offer lies not in the control of the three dimensional geometry,

but in the ability to actively and independently control the three dimensional mechanical properties of a structure with a given geometry. We can actively and seamlessly transition between anisotropic and isotropic stiffness and yield matrices, by jamming along different directions. In traditional jamming structures there is only a single jammed state available where all units are isotropically jammed (for vacuum based structures), or jammed along a single dimension (for electrostatic-clutch like jamming structures). We hope that this clarifies the confusion.

Traditional vacuum-based granular-jamming grippers also do not have active and programmed shape-morphing capabilities, they are able to passively deform under external loads, and conform to various shapes. In light of this comment, however, we have decided to include additional demonstrations to highlight the unique potentials untethered jamming introduces, especially in more random and complex interactions with its environment. Since the subunits can be fully independent from each other (compared to all other common jamming actuation methods, including vacuum), the jamming structure does not have to be one single cohesive body, it can be divided and distributed around the environment and foreign objects. Please refer to Figure 5c and Supplementary Video SV10 for the demonstration of distributed jamming.

2. On Page 14, the authors analyze the oscillatory behavior observed in sliding mode. This oscillatory problem could be addressed by incorporating subunits with randomly distributed magnetic composites. The authors may provide optimized results demonstrating the most smoothed force-displacement curve the structure can achieve.

1.2 We thank the reviewer for this comment.

- a. The oscillations are not necessarily a “problem”. Discretizing motion is necessary for some applications. For example, in the case of robotic position control and path planning, the ability to represent the deformation/displacement in a finite set of discrete states, rather than continuous possibilities simplifies computation and allows for control with less processing power. We have modified the text to be more clear in this regard. (Lines 317-319)
- b. We have experimentally shown that the oscillatory behavior could be modified by changing the distribution of the magnetic components within the composites. The results are shown in Fig. 3e in the original manuscript. The sample used for these experiments not only had irregular and randomized spacing of pillars, but also differing sizes of pillars. Since this experimental result varies multiple parameters simultaneously, we have added additional simulation results to the supplementary information that allow for a more methodical approach to how different design parameters influence oscillatory behavior. Specifically:
 - (1) Creating a coprime relative distribution between neighboring subunits, to avoid cases where the two sides have fully matching or fully out of phase patterns, and ensure consistent overlap throughout deformation regime.
 - (2) Modifying relative diameters between neighboring subunits while keeping the spacing coprime.

These simulation results show that the design space is multidimensional and complex, since the pattern not only influences the oscillation of the attractive force between the units throughout the deformation, but also the magnetization of the individual subunits. Our additional results demonstrate methodical design strategies that can lay the foundations of multi-objective optimization frameworks. These can be found in Supplementary Information, specifically in Supplementary Fig. 6.

3. Some error bars are missing in figures, e.g., Figure 3i and j. The authors should include error bars in all relevant figures to provide comprehensive results.

1.3 We thank the reviewer for pointing this out, and have modified the figures accordingly.

Reviewer #2 (Remarks to the Author):

This article titled “Jamming with magnetic composites” studies an externally controllable and tunable jamming phenomenon of centimeter-scale units. The jamming, or shape-locking, could be achieved along one dimension, in a plane, or in a volume. One difference between the proposed jamming strategy with other existing ones is that it is untethered and wirelessly triggered using a magnetic field. The working principle of this jamming strategy is based on the magnetic properties of soft ferromagnetic particles with low remanence and low coercivity. When an external magnetic field is applied, these magnetic particles are magnetized and they generate attractive forces between each subunit, causing jamming. When the magnetic field is removed, these particles do not retain much magnetization and thus the attractive force disappears, leading to unjamming. This article also characterized the force-deformation behaviors of the jamming structures using analytical models, numerical studies, and experimental validations. The article mentioned that the targeted applications of the proposed jamming mechanism include surgical robotics, micromanipulation, and structural monitoring. The reviewer has the following questions and comments.

1. The main novelty of this work is an untethered jamming mechanism. This article compares the proposed jamming mechanisms with some existing ones, especially the one applies a vacuum to the aggregate with an airtight envelope and the one utilizes a voltage. Then this article claims the key advantage of the proposed one is that it is untethered and easy to scale and reconfigure. However, there lacks a comparison between the proposed jamming mechanism with existing untethered ones. For example, there are previous studies on magnetically induced jamming. Here are some references:

- a) Khalil ur Rehman et al., Magnetic microactuators based on particle jamming, ACS Materials Letters 2024.*
- b) T Leps et al., A low-power, jamming, magnetorheological valve using electropermanent magnets suitable for distributed control in soft robots. 2020.*
- c) Xianhu Liu et al., Magnetic field-driven particle assembly and jamming for bistable memory and response plasticity, Science Advances, 2022.*
- d) Xianhu Liu, Magnetic field-induced particle assembly and jamming, Doctoral thesis, Aalto University, Finland. 2024.*

2.1 We thank the reviewer for their kind reminder and have now cited these references in the revised manuscript. Indeed, these studies are related to magnetic actuation and jamming; however, we would like to highlight that our work is fundamentally different from these in various ways.

- a. All of these references utilize homogeneous spherical magnetic particles. This results in two key differences:
 - The homogeneous subunits are not capable of directional controllability of the jamming behavior. They have no pre-programmable easy axes that determine the possible direction(s) of jamming. Our work focuses on the design and creation of specific subunits with the ability to achieve multi-dimensional and programmable directional jamming behavior. Composite design also enables the consideration of the repulsion forces induced between subunits.
 - In the jamming of spherical particles, a kinematic deformation mode dominates, where the jamming is a result of the kinematic locking between the constituents based on rearrangement. In all these prior studies, the magnetic attraction between subunits is only a portion of the jamming behavior and it is coupled with the rearrangement of subunits that find energetically favorable arrangements. Our work allows for a focus on the jamming induced purely by magnetic attraction forces.
- b. The jamming mechanism in these prior works operate in a specific confined space that induces varying boundary conditions alongside the magnetic fields, which have an influence on the force chains formed.
 - For example, in papers (c and d), the jamming structures are formed on a flat plate, and the magnetic field is applied asymmetrically from underneath this plate. This would induce magnetic field gradients that pull the particles towards the plate, contributing to

the jamming force chains. This is actually beneficial for the applications they are concerned with, such as inducing conductive connections between two parallel plates. However, there are also applications in which a jamming structure needs to operate in a free-standing manner. With our experiments such as lifting experiments in Supplementary Video SV2, we look at free-standing jamming structures. The jamming force chains are along a free-standing direction, without the effect of boundary conditions. These demonstrate how the jamming of subunits can be controlled in a fully untethered way, without reliance on boundary conditions.

- In paper (b), the jamming is utilized for a valve, the subunits are in a confined space within the valve and the electromagnetic actuation mechanism is directly integrated into the valve structure, which needs to be tethered to an external power source.
- c. And lastly, we kindly disagree with the reviewer's point about paper (a). The work by Rehman et al. demonstrates how the bulk magnetic response of a composite (made of magnetic and non-magnetic particles within a porous polymer), could be modified by jamming the aggregate through the swelling of the polymer. The jamming is not induced magnetically, but through swelling. However, we agree that this work is relevant and have referenced it in our discussion. (Lines 647-657)

2. Another key novelty in this article is the ability to selectively jam structures along specific directions. Is it the case that one specific kind of subunits can only be jammed along a predefined direction and cannot be altered after fabrication? It is hinted in the article that the proposed jamming mechanism can overcome the limitation of only having one "easy axis". But based on the presented results, it seems like that one type of subunits can only be jammed along one direction. If another direction is desired, a different type of subunits need to be deployed. It is obvious the jamming direction can be programmed by varying the design parameters. But once the subunits have been fabricated, i.e., all the parameters are fixed, can this jamming direction still be altered on-the-fly?

2.2 The subunits can be jammed along multiple directions after fabrication, this is the main contribution we have highlighted in Figure 4. The proposed 2-D structures can be jammed along two independent axes or both axes simultaneously, on-the-fly. Similarly, the 3-D structures can be jammed selectively in all combinations of the three main axes. Figure 4 and Supplementary Videos (SV) 5, 6, and 7 are all demonstrating this. In Fig. 4a and SV5, it can be seen that the same exact subunits in a 3-by-3 grid can be jammed along x, y, or both dimensions simultaneously, based on the magnetic field direction applied. In Fig. 4b and SV6, a 3 dimensional structure is demonstrated, where the exact same 6 subunits can be jammed selectively along different dimensions (only x, only y, only z, xy, xz, yz, or xyz), based on the direction of the applied magnetic field. In Fig. 4c and SV7, a 2-D structure is jammed selectively in multiple dimensions. We hope that this clarifies the reviewer's concerns.

3. This article states that an important motivation and application of this jamming mechanism is to assist robots in hard-to-reach space inside human body. But the prototypes presented in the current manuscript are at the scale of tens of centimeters. They are way too large to enter human body. And there aren't any straightforward methods or clear demonstrations that these prototypes could be downscaled by one or two orders of magnitude.

2.3 We understand the concern and appreciate the feedback. We also would like to mention that the response to this question has parallels with the response to point 1.1a above in response to Reviewer 1, and we have replicated some of the text in case the reviewers don't have access to all responses.

- a. First, we would like to clarify the scale of the experimental data we have provided. The samples are not in tens of centimeters. The subunits sizes range from samples from 3 mm side-length cubes to 8 mm side-length cubes. The 16 mm length cubes were only used for the characterization experiments looking into the different modes of interaction between two components. The goal was to have higher magnetic interaction forces such that we have better signal to noise ratio in our experimental data. The largest multi-subunit jamming structure we have had in the publication has had a dimension of 5 cm. These types of structures would actually get more challenging to actuate if they were at the scale of tens of centimeters, as controlling the magnetic field in such a large space would require magnetic navigation systems much larger than the one which was utilized in this study.

- b. To address the concern about applicability, we have added more explicit demonstrations illustrating specific functionalities for applications including but not limited to medical robotics. Specifically, in our new section titled “Interaction with Objects and Environment”, we have added:
- A magnetic jamming-based gripper that can apply a range of pinching/grasping forces based on the applied magnetic field. (See Fig. 5a and Supplementary Video SV8)
 - A dilation beam that can be deployed while flexible, and can gradually anchor itself and apply force to soft environments with the application of increasing magnetic fields. (See Fig. 5b and Supplementary Video SV9)

In the context of biomedical applications, procedures such as endoscopy or colonoscopy would be a good fit for these new demonstrations. Endoscopes and colonoscopes can have diameters up to 12 mm and, have working channel diameters up to 4 millimeters, which means they can easily deliver tools up to this size. Additionally, wireless capsule endoscopes typically have a size of 11 mm × 26 mm, as described in a review article published in Nature Communications in 2024 (1). The types of jamming structures proposed here could either be integrated into these capsule endoscopes or be delivered with endoscopes or colonoscopes. In light of these types of applications, we have chosen to create structures with 3-4 mm sized subunits, which are in a similar size-range to a number of granular jamming studies that also incorporate non-standard subunit designs (2, 3). This size also allows for the easier retrieval of the subunits, as they aren't so small that it becomes challenging to visualize with common medical imaging modalities. We would like to note that these structures are smaller than other magnetically actuated reconfigurable surgical tools in literature such as (4).

- c. The scale we have focused on for this paper is on the order of millimeters, not only due to the application spaces we were interested in, but also to have a design and fabrication platform accessible to researchers at various institutions with diverse sets of resources. However, there are microfabrication methods to scale down these structures to the order of micrometers, utilizing existing microfabrication methods. The 1D structures could be fabricated with a combination of standard photolithography and electroplating. There are studies which have developed fabrication protocols for similar morphologies (5–7). 2D and 3D structures are a bit more complex, Two-photon lithography would be one method to utilize for this purpose, as previously demonstrated in studies such as (8, 9).

4. An important limitation of the proposed jamming mechanism is that it requires continuous power consumption to maintain the applied external magnetic field. How does the proposed strategy compare with existing jamming mechanisms from the energy consumption perspective? One of the mentioned applications is untethered robots in minimally invasive healthcare, many of which rely on magnetic actuation. How does this jamming mechanism work alongside those magnetically controlled robots? Will they be compatible in the same workspace? Or maybe this jamming mechanism can only work with robots that use a non-magnetic working principle? It is an important point to clarify.

2.4 There are many excellent points within this comment, and we thank the reviewer for bringing all of them up. We have added some high-level comments in the main text, but have a more exhaustive version here:

- a. Regarding energy consumption:
- In the context of medical applications safety considerations are prioritized over energy consumption considerations. In this case, the “default” and “off” state of the mechanism should be designed considering the specific robotic task and context. For example, for devices which can actively apply high-forces to tissues around them at the jammed state, we would want the structure to unjam and soften when there are system-wide issues. In such cases, requiring the jamming mechanism to only be “stiff” and “high-force” when there is an active magnetic field applied makes more sense. This was our main consideration. However, there could be contexts where a structure would be “safer” when it is in its jammed state, for example when conforming to the tissues around is not wanted, or if the structure is trying to remain anchored in a part of the body against the

flow, etc. For structures in such a context, then considering an alternative “default” and “off” state is necessary.

- There are multiple strategies to change what the “default” state might look like, based on the main mechanism shared in this publication:
 - (1) The soft magnetic components used in the composite design can be chosen from a material that has a high-enough magnetic remanence to remain somewhat jammed when an external magnetic field is removed. This material would still have low coercivity, such that it is easy to unjam or jam the structure further.
 - (2) Permanent magnets could be directly embedded into the design of the jamming structure, ensuring a locally applied magnetic field, rather than relying on an external magnetic manipulation system. The jamming behavior can then be further modified with the externally applied magnetic field.
 - (3) Mechanical metamaterial design features could be incorporated into the design itself to induce interlocking behaviors between individual subunits to maintain in the jammed state (e.g. velcro-like surface).
- b. Regarding the point about integrating the jamming structure into existing magnetically actuated systems for minimally invasive medicine, and whether our jamming mechanism is compatible with such systems: This is actually quite straightforward, and was one of the main motivations for the design approach. While this work is mainly about the fundamentals of magnetic jamming, we believe that it is especially relevant and timely for medical applications with the increasing interest and demand in magnetically controlled medical procedures. A number of the clinically-ready magnetic manipulation systems in literature are able to control magnetic field magnitudes, directions, and gradients independently in all three dimensions. All of these enable multiple degrees of freedom. The jamming actuation we have proposed is controlled mainly by the magnitude of the applied magnetic field and the direction. The control of typical magnetic medical robots; however, rely on other aspects of the applied magnetic field. While there are many different possibilities, we outline here the most common ones:
 - A magnetic field gradient is utilized to create a pulling force in a desired direction. This is a completely different degree of freedom. In this case, the jamming mechanism could be directly incorporated into existing robotic systems without interfering with the actuation based on gradient. (This can be seen in action in our updated manuscript as well, in the new results we have added, specifically with magnetic jamming based gripper demonstration, Figure 5a and Supplementary Video SV8)
 - There are also magnetically actuated robots that are moved through the environment by utilizing a rotational movement (e.g. locomotion through tumbling), by applying a rotating magnetic field. Our 1-D jammed structures would be very straightforward to integrate into these systems, as they also have a single easy axis which a torque can be applied to through a rotating magnetic field. The ability to rotate jammed systems can also be seen in our new demonstration on distributed jamming. (This is especially visible in Supplementary Video SV10)

5. This article claims that the jamming is reversible (page 6 line 101). But all the demonstrations, especially the ones shown in the supplementary videos, do not contain reversible jamming and unjamming. For example, in supplementary video 1, the 1-D, 2-D, and 3-D tests start with an already jammed state and become unjammed when the magnetic field is removed. Will they be jammed again if the magnetic field is applied afterwards? It looks like the jamming state can only be achieved when all the subunits are well aligned to start with. In other words, the subunits may need to be manually placed into a certain pattern before applying a magnetic field to induce jamming. If that is the case, then this jamming-unjamming transition is not reversible and this point should be elaborated in the manuscript.

2.5 We thank the reviewer for this comment, since reversibility is very important for jamming structures for robotics applications. The reviewer is correct that in SV1, we only show the unjamming behavior. SV2, SV5, and SV 6, on the other hand, demonstrates the jamming behavior. SV7

demonstrates the jamming and unjamming behavior repeatedly. Therefore, we have already demonstrated that the jamming and unjamming behavior is reversible. Our additional experiments and supplementary videos (SV8, SV9, and SV10) also further demonstrate this reversibility.

Additionally, the reviewer has pointed out that the subunits “may need to be manually placed in a certain pattern”. This is true, if we want to achieve a deterministic morphology. And this was indeed the main approach in the paper, since the focus is on creating tunable isotropic/anisotropic mechanical properties of a structure with a given geometry, and not on the generation of magnetic assemblies by applying external magnetic fields. In light of this comment as well as a comment made by Reviewer 1, however, we have decided to include additional demonstrations to highlight the unique potentials untethered jamming introduces, especially in more random and complex interactions with the environment. Since the subunits can be fully independent from each other, the jamming structure does not have to be one single cohesive body, it can be divided and distributed around the environment and foreign objects. Please refer to Figure 5c and Supplementary Video SV10 for the demonstration of distributed jamming. In these videos, it can be seen that the jamming behavior depends on the subunits’ relative positions, and the behavior is much more stochastic. Additionally, it can also be seen that the jamming and unjamming transition is reversible.

6. Page 5 - Line 91-94: *“The effective susceptibility of the subunits (χ_{eff}) can be determined experimentally or numerically by $H = M \chi_{eff}$, where M is the resulting magnetization of the subunit, and H is the applied external field. [14]” This equation is wrong. It is unclear if this error is just a simple typo, or this wrong equation is used in the development of the analytical model and numerical calculations. Since this mistake is repeated in Methods - Numerical simulations (page 29 line 637), the reviewer is worried that this mistake is probably not just a typo and could have caused a systemic miscalculation in the analytic and numerical models.*

2.6 This is indeed a typo in the manuscript, and we thank the reviewer for noticing. The analytic and numerical models have the correct equation in the models, and we have corrected the text in the manuscript.

7. *What are the soft-ferromagnetic materials exactly? It is not mentioned in the Methods section or SI.*

2.7 They are iron based magnetic materials with a magnetically soft material, such as Ni, Fe, and Fe₃O₄. In our case we used 304 stainless steel wires. We have added this information to the methods section.

8. *It lacks obvious applications of the proposed jamming mechanism. Lots of potential applications are discussed in the section of Discussion. But none of these mentioned applications are demonstrated or can be directly inferred intuitively from the demonstrations presented in the current manuscript.*

2.8 We have added two new demonstrations to help connect to potential applications. These are already integrated into the comment 2.3 above, and can be found in the new section in the paper titled Interaction with Objects and Environment.

Reviewer #3 Comments

The authors present an untethered jamming mechanism based on magneto-mechanical coupling in soft-ferromagnetic composites. This approach has the potential to address key limitations of existing jamming actuation methods, which often require tethering, making structures challenging to scale and reconfigure. The study establishes design guidelines for programming the magnetization of subunits—the fundamental building blocks of the magneto-mechanical jamming system—to enhance performance. Analytical and numerical studies provide a framework for predicting, programming, and actively controlling the mechanical properties of these magnetic composite subunits. The authors further demonstrate the versatility of the mechanism through multi-dimensional jamming in 1-D, 2-D, and 3-D structures, including the ability to selectively jam along specific directions.

Jamming refers to the phenomenon where an aggregate transitions from a fluid-like state to a solid-like state. While extensively studied, existing jamming actuation methods, such as those using vacuum or voltage, lack untethered tunability and reconfigurability. The main contribution of this study is the introduction of an untethered jamming mechanism enabled by remote magnetic actuation. Overall, this work is both compelling and impactful. The article presents contributions, is well-organized, and mostly well-prepared. I believe the findings certainly deserve publication in Nature Communications and will be of considerable interest to researchers in the field of jamming and robotics. I recommend the paper for publication following adequate responses to the comments outlined below.

3.0 We thank the reviewer for their positive and encouraging feedback.

The attractive force between neighboring subunits plays a pivotal role in the proposed untethered jamming mechanism. Although the experimentally measured attractive force between neighboring subunits agrees with the FEM simulation results in the overall trend (see Fig. 2b), indicating that the jamming force can be remotely tuned, there are still considerable differences in specific values between the experimental tests and the FEM simulation results. The authors might comment on this point.

In addition, as mentioned by the authors in the section “Magnetically induced mechanical behavior”, the attraction force between two cubes, if approximated as two soft-magnetic dipoles, can be expressed as $F_{\text{attraction}} = \frac{\mu_0 H^2 (V \chi_{\text{eff}})^2}{4\pi d^4}$. Can the author use this analytical estimation to predict the attractive force between neighboring subunits and compare the results with the experimental and FEM simulation results?

3.1 We have incorporated new experimental results with an improved experimental setup that have a better match between experimental tests and FEM simulations. These can be found in Fig. 2b. In the measurements we had made in the original manuscript, we had used a manually controlled micrometer-scale translation stage. We suspected that the error might have come from this, since the distance between two magnetic bodies greatly influences the attraction force. We re-did the experiments with a motorized stage. We also improved our “zeroing” procedure, where we let the neighboring subunits self-assemble under the applied magnetic field and used that as the “zero” starting point before we started pulling the subunits apart. It can be seen that these modifications have greatly improved our results.

We have also added analytical estimations, based on the dipole-dipole model to the plot in Fig. 2b. The effective magnetization values used for the analytical model were extracted from simulations. We expect designers to also take a similar approach since analytical calculations become challenging when there are multi-pillar magnetic interactions at play.

There are still some discrepancies between FEM simulations, analytical models, and experimental results, and these come from:

- a. Fabrication errors: the cubes are manually sanded after waterjet cutting. This leads to variations in not just overall length of the pillars, but also in the surface roughness along the contact surface between the subunits. This influences both the zeroing procedure, as well as introducing slight variations in alignment.

- b. At larger fields, such as 25 and 30 mT, our magnetization experiments have shown that the materials we used start to saturate. The finite element models and the analytical models do not take into account this saturation, and assume a constant magnetic susceptibility for all field magnitudes. Therefore, the mismatches at these higher fields are expected.

The reviewer is not clear about how the deviation of magnetization direction (as presented in Fig. 2h-i) is defined. Here free-standing subunits are considered, and an external magnetic field is applied to magnetize them. As far as I understand, the magnetic phase in the subunits will be magnetized along the direction of the applied magnetic field. How is the deviation of magnetization direction formed? Is this deviation angle a volumetrically averaged value?

3.2 We thank the reviewer for this comment. We were actually not considering free-standing subunits when looking into the magnetization direction in the FEM models. In the simulation, they were fixed in space (they weren't allowed to rotate), and the magnetic field was applied in one direction. Then, the subunit magnetizes not directly along the axis but has a slight deviation, due to the geometry. The deviation angle is calculated by a volumetrically averaged value. To understand why there is any deviation, it might help to consider the perfect case: an ellipsoid. If the subunits were a homogeneous magnetic material in the shape of an ellipsoid, with the magnetic field applied along its longer dimension, then we would get a "perfect" easy axis, with no deviation. For cases like a square prism, there are actually multiple easy axes along the different diagonals. Here, the goal was to see how "easy" our "easy-axis" actually is, and what parameters this depends on. We added clarification in the text as well to make this point more clear. (Lines 247-251)

Regarding the pivoting mode illustrated in Fig. 3f, the subunit will rotate about the pivot point after the breakaway. With larger deformations, the subunit's easy axes will significantly deviate from the direction of the applied magnetic field. In this case, will the subunit be further magnetized along the direction of the applied magnetic field (instead of its easy axis direction)?

3.3 We appreciate this important insight and question. It is absolutely correct that the direction of the applied magnetic field influences the direction of magnetization. However, this direction is also highly (and predominantly) dependent on the easy axis of the subunit. If there is a misalignment between this axis and the externally applied magnetic field, the "easier" the easy axis is, the closer the resulting magnetization direction will be to the easy axis. While we expected the magnetization direction to not deviate significantly from the easy axis due to our composite design, we added simulation results to quantify this deviation. As it can be seen from Fig. R1, as the subunit pivots with respect to the magnetic field, at small angles, the magnetization is still along the easy axis of the subunit. (The black line is the "ideal" situation, if the magnetization direction were always perfectly aligned with the easy axis, despite the pivoting angle). At larger angles, however, a deviation from the easy axis is observed. There is a linear trend of increasing deviation that reaches almost 2.5 degrees when the pivot angle is 30 degrees. This percentage of error is within the bounds of errors generally expected between magnetization behaviors in experiments vs. simulations. Additionally, at these larger angles, since the subunits are further from each other, the attraction force between subunits is minimal, and the alignment torque dominates the behavior, so this deviation becomes less relevant.

Figure. R1. Magnetization direction as a function of pivoting angle (Black: ideal, Red dots: simulation).

Minor points:

In line 303 of page 15, the local internal tensile force should be F_x , and the shear torque should be T_x , according to the notations in Fig. 3h.

The point load in Fig. 3i-j is denoted as F , while in the main text it is represented as P .

In the section “Multi-directional jamming”, the cartesian axes are labeled with lowercase x , y , and z in some places and uppercase X , Y , and Z in other places.

3.4 We thank the reviewer for noticing these points and have fixed them.

References

1. Q. Cao, R. Deng, Y. Pan, R. Liu, Y. Chen, G. Gong, J. Zou, H. Yang, D. Han, Robotic wireless capsule endoscopy: recent advances and upcoming technologies. *Nat. Commun.* **15**, 4597 (2024).
2. A. G. Athanassiadis, M. Z. Miskin, P. Kaplan, N. Rodenberg, S. H. Lee, J. Merritt, E. Brown, J. Amend, H. Lipson, H. M. Jaeger, Particle shape effects on the stress response of granular packings. *Soft Matter* **10**, 48–59 (2014).
3. D. Howard, J. O’Connor, J. Brett, G. W. Delaney, “Shape, Size, and Fabrication Effects in 3D Printed Granular Jamming Grippers” in *2021 IEEE 4th International Conference on Soft Robotics (RoboSoft)* (IEEE, New Haven, CT, USA, 2021; <https://ieeexplore.ieee.org/document/9479438/>), pp. 458–464.
4. H. Gu, M. Möckli, C. Ehmke, M. Kim, M. Wieland, S. Moser, C. Bechinger, Q. Boehler, B. J. Nelson, Self-folding soft-robotic chains with reconfigurable shapes and functionalities. *Nat. Commun.* **14**, 1263 (2023).
5. H. Gu, E. Hanedan, Q. Boehler, T.-Y. Huang, A. J. T. M. Mathijssen, B. J. Nelson, Artificial microtubules for rapid and collective transport of magnetic microcargoes. *Nat. Mach. Intell.* **4**, 678–684 (2022).
6. L. Amato, S. S. Keller, A. Heiskanen, M. Dimaki, J. Emnéus, A. Boisen, M. Tenje, Fabrication of high-aspect ratio SU-8 micropillar arrays. *Microelectron. Eng.* **98**, 483–487 (2012).
7. J. D. Williams, Study on the postbaking process and the effects on UV lithography of high aspect ratio SU-8 microstructures. *J. MicroNanolithography MEMS MOEMS* **3**, 563 (2004).
8. C. C. J. Alcântara, F. C. Landers, S. Kim, C. De Marco, D. Ahmed, B. J. Nelson, S. Pané, Mechanically interlocked 3D multi-material micromachines. *Nat. Commun.* **11**, 5957 (2020).
9. A. Vyatskikh, S. Delalande, A. Kudo, X. Zhang, C. M. Portela, J. R. Greer, Additive manufacturing of 3D nano-architected metals. *Nat. Commun.* **9**, 593 (2018).

Round 1 reviewer's comment

Round 1 authors' responses

Round 2 reviewer's comment

Round 2 authors' responses

Reviewer #1

I have reviewed the revised manuscript entitled “Jamming with Magnetic Composites” and the authors’ responses to the initial comments. I appreciate the thorough efforts in addressing all the concerns raised during my first review. The revisions, including the innovation concerns compared with the vacuum-based jamming techniques and oscillations phenomenon, have significantly improved the clarity and quality of the manuscript. The key points have been adequately addressed, and this work is now ready for publication. Therefore, I recommend that the manuscript be accepted for publication.

1.0 We are pleased to hear that our revisions have addressed all concerns raised during the first revision. We would once again like to thank the reviewer for their thoughtful questions and suggestions, we strongly believe their contributions have improved our manuscript’s quality and impact.

Reviewer #2

The reviewer thanks the authors for the clarifications, explanations, and added experiments. Unfortunately, the reviewer thinks the critical problems of this work remain.

2.0 We thank the reviewer for their additional thoughts and suggestions. While we have addressed the comments point by point below, we would like to clarify what seems to be an overarching misunderstanding, specifically regarding the main contributions of the paper. The manuscript is focused on achieving, modeling, and validating untethered jamming structures enabled by magnetic composites. Possible application areas, based on prior work and the magneto-mechanical models in the publication are mentioned (such as surgical robotics and micromanipulation), yet are not the main focus. The demonstrations included in the manuscript are intended as proof-of-concept examples to illustrate the advantages of magnetic jamming structures compared to other jamming structures to date. We emphasize that, unlike existing jamming actuation methods such as vacuum- or voltage-based systems—which typically require tethering through tubing or wiring, or rely on bulky on-board components like pumps or batteries—our magnetic jamming approach enables untethered, compact actuation. This capability is particularly advantageous for applications in constrained or hard-to-access environments, such as within the human body. To avoid any potential confusion, we have revised the manuscript to clarify that the primary contribution lies in the development and validation of the untethered magnetic jamming mechanism, rather than in demonstrating specific end-use applications.

This article titled “Jamming with magnetic composites” studies an externally controllable and tunable jamming phenomenon of centimeter-scale units. The jamming, or shape-locking, could be achieved along one dimension, in a plane, or in a volume. One difference between the proposed jamming strategy with other existing ones is that it is untethered and wirelessly triggered using a magnetic field. The working principle of this jamming strategy is based on the magnetic properties of soft ferromagnetic particles with low remanence and low coercivity.

When an external magnetic field is applied, these magnetic particles are magnetized and they

generate attractive forces between each subunit, causing jamming. When the magnetic field is removed, these particles do not retain much magnetization and thus the attractive force disappears, leading to unjamming. This article also characterized the force-deformation behaviors of the jamming structures using analytical models, numerical studies, and experimental validations. The article mentioned that the targeted applications of the proposed jamming mechanism include surgical robotics, micromanipulation, and structural monitoring. The reviewer has the following questions and comments.

1. The main novelty of this work is an untethered jamming mechanism. This article compares the proposed jamming mechanisms with some existing ones, especially the one applies a vacuum to the aggregate with an airtight envelope and the one utilizes a voltage. Then this article claims the key advantage of the proposed one is that it is untethered and easy to scale and reconfigure. However, there lacks a comparison between the proposed jamming mechanism with existing untethered ones. For example, there are previous studies on magnetically induced jamming. Here are some references:

a) Khalil ur Rehman et al., *Magnetic microactuators based on particle jamming*, *ACS Materials Letters* 2024.

b) T Leps et al., *A low-power, jamming, magnetorheological valve using electropermanent magnets suitable for distributed control in soft robots*. 2020.

c) Xianhu Liu et al., *Magnetic field-driven particle assembly and jamming for bistable memory and response plasticity*, *Science Advances*, 2022.

d) Xianhu Liu, *Magnetic field-induced particle assembly and jamming*, *Doctoral thesis*, Aalto University, Finland. 2024.

2.1 We thank the reviewer for their kind reminder and have now cited these references in the revised manuscript. Indeed, these studies are related to magnetic actuation and jamming; however, we would like to highlight that our work is fundamentally different from these in various ways.

- a. All of these references utilize homogeneous spherical magnetic particles. This results in two key differences:
 - i. The homogeneous subunits are not capable of directional controllability of the jamming behavior. They have no pre-programmable easy axes that determine the possible direction(s) of jamming. Our work focuses on the design and creation of specific subunits with the ability to achieve multi-dimensional and programmable directional jamming behavior. Composite design also enables the consideration of the repulsion forces induced between subunits.
 - ii. In the jamming of spherical particles, a kinematic deformation mode dominates, where the jamming is a result of the kinematic locking between the constituents based on rearrangement. In all these prior studies, the magnetic attraction between subunits is only a portion of the jamming behavior and it is coupled with the rearrangement of subunits that find energetically favorable arrangements. Our work allows for a focus on the jamming induced purely by magnetic attraction forces.
- b. The jamming mechanism in these prior works operate in a specific confined space that induces varying boundary conditions alongside the magnetic fields, which have an influence on the force chains formed.
 - i. For example, in papers (c and d), the jamming structures are formed on a flat plate, and the magnetic field is applied asymmetrically from underneath this plate. This would induce magnetic field gradients that pull the particles towards the plate,

contributing to the jamming force chains. This is actually beneficial for the applications they are concerned with, such as inducing conductive connections between two parallel plates. However, there are also applications in which a jamming structure needs to operate in a free-standing manner. With our experiments such as lifting experiments in Supplementary Video SV2, we look at free-standing jamming structures. The jamming force chains are along a free-standing direction, without the effect of boundary conditions. These demonstrate how the jamming of subunits can be controlled in a fully untethered way, without reliance on boundary conditions.

- ii. In paper (b), the jamming is utilized for a valve, the subunits are in a confined space within the valve and the electromagnetic actuation mechanism is directly integrated into the valve structure, which needs to be tethered to an external power source.
- c. And lastly, we kindly disagree with the reviewer's point about paper (a). The work by Rehman *et al.* demonstrates how the bulk magnetic response of a composite (made of magnetic and non-magnetic particles within a porous polymer), could be modified by jamming the aggregate through the swelling of the polymer. The jamming is not induced magnetically, but through swelling. However, we agree that this work is relevant and have referenced it in our discussion.

The reviewer thanks the authors for the explanations.

We are pleased to hear that our response has satisfactorily addressed the reviewer's concern.

2. Another key novelty in this article is the ability to selectively jam structures along specific directions. Is it the case that one specific kind of subunits can only be jammed along a predefined direction and cannot be altered after fabrication? It is hinted in the article that the proposed jamming mechanism can overcome the limitation of only having one "easy axis". But based on the presented results, it seems like that one type of subunits can only be jammed along one direction. If another direction is desired, a different type of subunits need to be deployed. It is obvious the jamming direction can be programmed by varying the design parameters. But once the subunits have been fabricated, i.e., all the parameters are fixed, can this jamming direction still be altered on-the-fly?

2.2 The subunits can be jammed along multiple directions after fabrication, this is the main contribution we have highlighted in Figure 4. The proposed 2-D structures can be jammed along two independent axes or both axes simultaneously, on-the-fly. Similarly, the 3-D structures can be jammed selectively in all combinations of the three main axes. Figure 4 and Supplementary Videos (SV) 5, 6, and 7 are all demonstrating this. In Fig. 4a and SV5, it can be seen that the same exact subunits in a 3-by-3 grid can be jammed along x, y, or both dimensions simultaneously, based on the magnetic field direction applied. In Fig. 4b and SV6, a 3 dimensional structure is demonstrated, where the exact same 6 subunits can be jammed selectively along different dimensions (only x, only y, only z, xy, xz, yz, or xyz), based on the direction of the applied magnetic field. In Fig. 4c and SV7, a 2-D structure is jammed selectively in multiple dimensions. We hope that this clarifies the reviewer's concerns.

The reviewer thanks the author for the response. The reviewer apologizes for not describing this question clearly in the previous round of peer review. The reviewer would like to further explain what the question is. It is stated in the manuscript (both the original and the revised one) that

"Additionally, the shape-dependent magnetization behavior of soft-ferromagnetic materials was leveraged for designing subunits. Specifically, a continuous soft-magnetic body has an "easy axis", a direction along which it is more energetically favorable to magnetize,

typically the more elongated dimension of the body^{15,16}. If a jamming structure's subunits are made of a soft-ferromagnetic material, jamming will only occur along the longer axis of the subunit and only in one dimension. This limits the possible morphologies and functionalities of the resulting jamming structures.

Magnetic composite subunits that have soft-ferromagnetic components within a non-magnetic matrix can help overcome these limitations. The magnetic constituents' easy axis can be aligned along the desired direction of jamming irrespective of the overall geometry of the subunits, enabling programmability of jamming directions."

This statement identifies the observation that jamming can only occur along one direction (the easy axis) as the limitation and promises the proposed strategy can overcome this limitation and enable the programmability of jamming directions. And the authors are correct that Fig. 4 shows that jamming of a 3-by-3 grid can happen along x, y, or both dimensions at the same time. But the reviewer's concern is that, for one specific subunit, its possible jamming direction(s) is fixed after fabrication. For example, the subunit utilized in Fig. 4 can only be jamming along x, y, or both directions, while the subunit in Fig. 1a can only be jammed along its thickness. The possible jamming directions are always discrete directions, i.e., we get to choose one from a limited number of possible directions. It is not a continuously variable jamming direction. That is why the reviewer suggests to properly define this "programmability", which is a very broad and strong claim.

2.2. We thank the reviewer for clarifying their original comment. Here, we aim to clarify the notion of "programmability" as presented in this manuscript, which encompasses more than what the current comment suggests.

Specifically, we identify **three physical properties** that can be programmed to achieve targeted jamming behaviors. Each of these properties includes:

- (a) **designable attributes** — values that are pre-programmed based on application needs, informed by our proposed models, and fixed during the fabrication process
- (b) **tunable attributes**, which can be dynamically adjusted in real-time through modulation of the applied magnetic field.

The two properties are as follows, followed by an explanation of the programmability our approach enables:

1. The inter sub-unit attraction force:

- a. The designable attribute here is the range of possible attraction forces, based on the magnetic susceptibility of the composite. We can "pre-program" the magnetic susceptibility with design, within the physical limits of the design constraints such as volume and morphology and considering the limitations our magnetic navigation system might have. We have already mentioned this as pre-programming in Line 111.
- b. The tunable attribute here is the actual attraction force between subunits, which changes the overall jamming behavior of a structure. This can be actively changed by altering the magnetic field's magnitude.

2. The directionality of jamming behavior:

- a. The designable attribute here is the possible directionalities once the sub-units are fabricated. Here we enable a shape-independent jamming possibility, which was not possible before our work. This has already been mentioned in Line 132, and we have added a clarifying sentence to this paragraph. A more nuanced aspect of this programmability also arises from our composite design, as highlighted through numerical models (Lines 225–237), which facilitate improved alignment of the

magnetization direction with the desired jamming axes, as it isn't trivial to get perfect alignment.

- b. The tunable attribute here is the direction of jamming which is enabled by the pre-programmed possibilities. This can be controlled by altering the direction of the applied magnetic field. This is the focus of the section on multi-dimensional jamming, starting in line 389. We have also added text here to further clarify some of the points.

Since this specific aspect of the programmability was the focus of this comment, we have also visualized this below in Fig R2.1. We have chosen to focus on Cartesian coordinate system focused directionalities since this is more generalizable, covers the three-dimensional space, and assists with robotic control applications that require different behavior in different degrees of freedom. However, different directionalities can also be chosen as shown in Figure R2.1.

3. The stick slip behavior in the sliding state:

- a. The designable attribute here is the wavelength of the oscillations, the inter-subunit distance difference in which the force oscillates. This can be pre-programmed based on the spacing and distribution of the pillars in the composite, as described starting at line 276.
- b. The tunable attribute is the amplitude of these oscillations, which can be changed by changing the magnetic field magnitude. With the current design of magnetic composites, this property is coupled with the control of the sliding force desired as well.

We hope that the distinction we have made here between designable and tunable attributes clarify the extent and modality of programmability we have focused on in this paper. Here, we would also like to highlight that there are many publications in literature that define programmability similarly, where there are discrete possibilities pre-programmed into a structure based on the specific resulting behavior required. Many of the pre-existing work on magnetic programmability focuses on achieving motion given specific magnetization profiles of hard (permanent) magnetic materials ^{1,2}. These inherently allow for behaviors that are geometrically more complex, as they focus on motion and shape change and utilize components that are pre-magnetized in different directions. **None of the systems in these works would allow for reversible jamming behavior, since the magnetic components remain magnetized. We thought highlighting this difference might also help distinguish our work, since our work focuses on tunable directional mechanical behavior (stiffness, yield) and not morphology or shape change.** Our proposed designs could be used to create more complex geometries that have tunable mechanical behavior, as shown below in Figure R2.2, however, we believe that this is outside the scope of the current manuscript and would dilute the main message, since the mechanical behaviors we have focused on (stiffness, yield) would be determined in the same way we have in the present manuscript.

What was possible before our magnetic composites:

The type of shape independent programmability enabled with our magnetic composites:

Figure R2.1 Programmability in terms of Directional Jamming Behavior

Figure R2.2 Combining different subunit designs can lead to more complex morphologies that also have on-the-fly tunability of stiffness and yield.

3. This article states that an important motivation and application of this jamming mechanism is to assist robots in hard-to-reach space inside human body. But the prototypes presented in the current manuscript are at the scale of tens of centimeters. They are way too large to enter human body. And there aren't any straightforward methods or clear demonstrations that these prototypes could be downscaled by one or two orders of magnitude.

2.3 We understand the concern and appreciate the feedback. We also would like to mention that the response to this question has parallels with the response to point 1.1a above in response to Reviewer 1, and we have replicated some of the text in case the reviewers don't have access to all responses.

- a. First, we would like to clarify the scale of the experimental data we have provided. The samples are not in tens of centimeters. The subunits sizes range from samples from 3 mm side-length cubes to 8 mm side-length cubes. The 16 mm length cubes were only used for the characterization experiments looking into the different modes of interaction between two components. The goal was to have higher magnetic interaction forces such that we have better signal to noise ratio in our experimental data. The largest multi-subunit jamming structure we have had in the publication has had a dimension of 5 cm. These types of structures would actually get more challenging to actuate if they were at the scale of tens of centimeters, as controlling the magnetic field in such a large space would require magnetic navigation systems much larger than the one which was utilized in this study.

The scale bars in the figures of the manuscript are 0.5 or 1 cm, indicating the reported devices are at least a few centimeters. Based on the demonstrations, at least three subunits are needed for any tasks and the volume taken is significant within a confined workspace.

2.3.a. We thank the reviewer for recognizing that the devices are not in “the scale of tens of centimeters” as they had stated in their original comment, but instead in the scale of a few centimeters maximum. We would like to address a few of the points here:

- The number of subunits required is a function of the task that is at hand. For example, if an application only needs force control through utilizing the sliding deformation mode, two subunits would be enough to achieve this, and would additionally reduce device complexity while increasing the possible force range. This was previously elaborated on in the previous response to reviewers (specifically a comment from Reviewer 1). We will quote this again here, since we think it might be helpful in this context:

Jamming structures do not generally need to have small unit cells. One of the great advantages of jamming structures for robotics applications is that they can be designed according to the performance requirements of the desired task. The dimensions of the subunits can be varied to program specific tunable mechanical behaviors, and more and smaller subunits per structure does not always mean better performance. These tradeoffs are different based on the actuation method utilized, and magnetic jamming has its own specific physical constraints. For example, in the case of the simple beam-bending we show in Figure 3i, the deformation is localized to one point due to the stress distributions in the beam. Having smaller but more pieces in this case would not change the resulting deformation, and it would actually diminish the overall jamming force achievable by the subunits, since they end up having less magnetic volume per-subunit. This is quite a different physical phenomenon compared to vacuum-based jamming, in which the actuation forces are generated externally, but not through the individual subunits themselves.

- With our newer experimental results (shown in Fig. 5a and Fig. 5b), as a response to reviewers' concerns and comments, we have demonstrated proof-of-concept jamming structures that could easily function (in terms of size) in the gastrointestinal tract. Improving the clinically relevant functionalities of these structures is beyond the scope of this current publication, which focuses on modeling and characterizing the fundamental magneto-mechanical behavior.
- b. To address the concern about applicability, we have added more explicit demonstrations illustrating specific functionalities for applications including but not limited to medical robotics. Specifically, in our new section titled "Interaction with Objects and Environment", we have added:
 - i. A magnetic jamming-based gripper that can apply a range of pinching/grasping forces based on the applied magnetic field. (See **Fig. 5a** and Supplementary Video SV8)

The reviewer appreciates the authors' effort to include more application-oriented demonstrations. But the presented gripper demonstration is over-simplified. There are similar (and more capable) demonstrations in the literature (E. Diller and M. Sitti, *Advanced Functional Materials* 2014). This added result doesn't really show what benefits or advantages the proposed strategy has over existing ones.

2.3.b.i. We completely agree that Prof. Eric Diller's work which has been cited by the reviewer offers much more functionalities with regards to gripping behaviors. However, we would like to remind the focus of the manuscript, which is on jamming, not on developing novel tools. Therefore, the goal of this demonstration is not at all to show a novel or better gripper design that has advantages over existing grippers. The goal of this specific demonstration is to utilize the soft magnetic composites capable of jamming in a familiar robotics example, in which the ability to jam, unjam, rotate, and move the jamming structure can be visibly shown, addressing the justified concerns of reviewers regarding whether jamming behavior could be controlled independent of other properties. Fabricating a novel gripper with complicated functionality is out of the scope of our manuscript.

- ii. A dilation beam that can be deployed while flexible, and can gradually anchor itself and apply force to soft environments with the application of increasing magnetic fields. (See **Fig. 5b** and Supplementary Video SV9)

In the context of biomedical applications, procedures such as endoscopy or colonoscopy would be a good fit for these new demonstrations. Endoscopes and colonoscopes can have diameters up to 12 mm and, have working channel diameters up to 4 millimeters, which means they can easily deliver tools up to this size. Additionally, wireless capsule endoscopes typically have a size of 11 mm × 26 mm, as described in a review article published in *Nature Communications* in 2024 (1). The types of jamming structures proposed here could either be integrated into these capsule endoscopes or be delivered with endoscopes or colonoscopes. In light of these types of applications, we have chosen to create structures with 3-4 mm sized subunits, which are in a similar size-range to a number of granular jamming studies that also incorporate non-standard subunit designs (2, 3). This size also allows for the easier retrieval of the subunits, as they aren't so small that it becomes challenging to visualize with common medical imaging modalities. We would like to note that these structures are smaller than other magnetically actuated reconfigurable surgical tools in literature such as (4).

The reviewer acknowledges that there are existing magnetically actuated surgical tools at the size scale of millimeter to centimeter. Specifically for the reference mentioned by the authors, i.e., H. Gu et al. *Nature Communications* 2023, this previous work has demonstrated more functionalities such as the integration of flexible PCBs and LEDs, programmable heating surface, integration with capsule endoscope, and integration with a medical grade thoracic catheter. Overall, the previous work presented a comprehensive investigation with its claims (including its size scale) well justified by experimental results. However, this manuscript lacks functionalities other than jamming, and how it could be integrated with medical devices remains unexplored.

2.3.b. We appreciate the reviewer reconsidering their size expectations for possible future biomedical application of magnetically actuated jamming structures, in light of literature and products we have mentioned in our previous response.

Here, we would like to reemphasize the focus of the manuscript. The main contribution is to achieve, model, and validate untethered and multidirectional magneto-mechanical jamming behavior. The focus is explicitly on jamming, and not on delivering a fully integrated multi-functional surgical system; therefore, other functionalities that are not jamming are not explored or highlighted. We agree that future work, such as creating electrical connections between the sub-units would enable further functionalities in terms of sensing, communication and actuation, but would like to emphasize that this is beyond the scope of the present paper. The present paper lays the magneto-mechanical foundations, and shows a few proof-of-concept demonstrators that demonstrates these functionalities at the relevant scales, enabling future translational work which incorporates integration with devices.

- c. The scale we have focused on for this paper is on the order of millimeters, not only due to the application spaces we were interested in, but also to have a design and fabrication platform accessible to researchers at various institutions with diverse sets of resources. However, there are microfabrication methods to scale down these structures to the order of micrometers, utilizing existing microfabrication methods. The 1D structures could be fabricated with a combination of standard photolithography and electroplating. There are

studies which have developed fabrication protocols for similar morphologies (5–7). 2D and 3D structures are a bit more complex, Two-photon lithography would be one method to utilize for this purpose, as previously demonstrated in studies such as (8, 9).

The reviewer agrees that several technologies exist for downscaling the reported devices. However, it isn't a straightforward path and many times it is even impracticable. This is the reason why some of the most influential work in magnetically controlled small-scale robots are investigations about fabrication, such as Y. Kim et al. *Nature* 2018 and J. Cui et al. *Nature* 2019. As a result, the reviewer can only evaluate the work based on what results have been actually presented, with an extension to what are also reasonably possible but not demonstrated. Unfortunately, downscaling the reported devices is beyond this.

2.3.c. We appreciate the reviewer's thoughtful comment. We fully agree that downscaling to the microscale is a non-trivial challenge, often requiring dedicated advances in fabrication techniques, as demonstrated by influential works such as Y. Kim et al (2018) and J. Cui et al. (2019). While we believe that downscaling is technically feasible through future development of suitable fabrication processes, such efforts are beyond the scope of the current study, which focuses on demonstrating the core mechanism and capabilities of the proposed system. Importantly, the system is already applicable to the targeted applications even at the current scale. To clarify this point, we have revised the relevant discussion in the manuscript to better reflect the limitations and realistic expectations regarding scalability.

Additionally, we would like to kindly reframe the relevance of the specific examples the reviewer has suggested.

- The Kim paper from 2018 focuses on the magnetization of a composite ink during 3D printing. Our work does not require pre-magnetization as we do not work with hard magnetic materials, but with soft magnetic materials. This simplifies the fabrication process dramatically, and as long as we can achieve multi-material printing with a magnetic and non-magnetic material we would be able to create our structures via 3D printing^{3,4}. We did not mention this formerly, as the comment had focused on downscaling. The resolution of the structures presented in the cited paper, as well as in other 3D printing focused fabrication methods, are actually similar to the resolution of the structures we have fabricated, since we have utilized wires similar in size to printer nozzle diameters. Therefore, this publication is not precisely relevant in the context of this specific comment.
- The Cui paper from 2019 focuses on the remagnetization of magnetic components in a continuous structure, again using hard magnetic materials. In order to be able to remotely remagnetize portions of the structure individually, hard magnetic materials with different hysteresis profiles are utilized throughout the structure. While the scale of the produced prototypes are relevant as they showcase the possibility of downscaling magnetic composites, the underlying control mechanisms and materials utilized are different, especially since we use soft magnetic materials which have minimal hysteresis.

If we would like to reference additional fabrication-focused publications that are aligned with the types of structures discussed in the present paper (in addition to the ones we have cited in our first response), it may be more appropriate to consider fabrication methods that focus not on magnetization during fabrication or remagnetization after fabrication but rather on creating anisotropic or specifically-aligned magnetic structures within a non-magnetic matrix^{5,6}.

4. An important limitation of the proposed jamming mechanism is that it requires continuous power consumption to maintain the applied external magnetic field. How does the proposed strategy compare with existing jamming mechanisms from the energy consumption perspective? One of the mentioned applications is untethered robots in minimally invasive healthcare, many of which rely on magnetic actuation. How does this jamming mechanism work alongside those magnetically controlled robots? Will they be compatible in the same workspace? Or maybe this jamming mechanism can only work with robots that use a non-magnetic working principle? It is an important point to clarify.

2.4 There are many excellent points within this comment, and we thank the reviewer for bringing all of them up. We have added some high-level comments in the main text, but have a more exhaustive version here:

a. Regarding energy consumption:

- i. In the context of medical applications safety considerations are prioritized over energy consumption considerations. In this case, the “default” and “off” state of the mechanism should be designed considering the specific robotic task and context. For example, for devices which can actively apply high-forces to tissues around them at the jammed state, we would want the structure to unjam and soften when there are system-wide issues. In such cases, requiring the jamming mechanism to only be “stiff” and “high-force” when there is an active magnetic field applied makes more sense. This was our main consideration. However, there could be contexts where a structure would be “safer” when it is in its jammed state, for example when conforming to the tissues around is not wanted, or if the structure is trying to remain anchored in a part of the body against the flow, etc. For structures in such a context, then considering an alternative “default” and “off” state is necessary.
- ii. There are multiple strategies to change what the “default” state might look like, based on the main mechanism shared in this publication:
 - 1) The soft magnetic components used in the composite design can be chosen from a material that has a high-enough magnetic remanence to remain somewhat jammed when an external magnetic field is removed. This material would still have low coercivity, such that it is easy to unjam or jam the structure further.
 - 2) Permanent magnets could be directly embedded into the design of the jamming structure, ensuring a locally applied magnetic field, rather than relying on an external magnetic manipulation system. The jamming behavior can then be further modified with the externally applied magnetic field.
 - 3) Mechanical metamaterial design features could be incorporated into the design itself to induce interlocking behaviors between individual subunits to maintain in the jammed state (e.g. velcro-like surface).

The reviewer thanks the authors for the explanation.

We are pleased to hear that our response has satisfactorily addressed the reviewer’s concerns.

- b. Regarding the point about integrating the jamming structure into existing magnetically actuated systems for minimally invasive medicine, and whether our jamming mechanism is compatible with such systems: This is actually quite straightforward, and was one of the main motivations for the design approach. While this work is mainly about the fundamentals of magnetic jamming, we believe that it is especially relevant and timely for medical applications with the increasing interest and demand in magnetically controlled

medical procedures. A number of the clinically-ready magnetic manipulation systems in literature are able to control magnetic field magnitudes, directions, and gradients independently in all three dimensions. All of these enable multiple degrees of freedom. The jamming actuation we have proposed is controlled mainly by the magnitude of the applied magnetic field and the direction. The control of typical magnetic medical robots; however, rely on other aspects of the applied magnetic field. While there are many different possibilities, we outline here the most common ones:

- i. A magnetic field gradient is utilized to create a pulling force in a desired direction. This is a completely different degree of freedom. In this case, the jamming mechanism could be directly incorporated into existing robotic systems without interfering with the actuation based on gradient. (This can be seen in action in our updated manuscript as well, in the new results we have added, specifically with magnetic jamming based gripper demonstration, Figure 5a and Supplementary Video SV8)
- ii. There are also magnetically actuated robots that are moved through the environment by utilizing a rotational movement (e.g. locomotion through tumbling), by applying a rotating magnetic field. Our 1-D jammed structures would be very straightforward to integrate into these systems, as they also have a single easy axis which a torque can be applied to through a rotating magnetic field. The ability to rotate jammed systems can also be seen in our new demonstration on distributed jamming. (This is especially visible in Supplementary Video SV10)

There are three most utilized degree-of-freedom of an exerted magnetic field for controlling such devices, i.e., the field strength, the field direction, and the field's variation in space and time. The proposed jamming mechanism requires two of them, i.e., the field direction and the field strength. It is true that we can still use the field's variation in space and time to control other parts independently. However, we should note that a magnetic field's spatiotemporal variance is heavily coupled with the field strength and direction. Whether we use an electromagnetic coil or a permanent magnet to create the magnetic field, we cannot control the field's spatiotemporal variance independently without also affecting its strength and direction. Unless the robot's main and only capability is based on this jamming (such as the gripper demonstration), this jamming mechanism is unlikely to be compatible with other magnetically controlled devices.

2.4.b. We completely agree with the reviewer that the multi-DOF actuation of jamming-based structures is a challenging (and exciting) control problem, specifically when the field strength and direction are considered.

We agree that the magnetic parameters: magnitude, direction, and gradient, are interdependent. However, active and independent control of these parameters is possible, as we have demonstrated in the gripper and distributed jamming example. Specifically:

- The gripper demonstration shows that a structure can be:
 - Translated and/or rotated without jamming (it can be seen that the field magnitude is kept constant enough, since the gripper does not close)
 - Jammed without motion (the gradual increase in gripping demonstrates the gradually increased magnetization)
 - Translated and/or rotated while jammed (it can be seen that the field is kept constant enough, since the gripper does not open)
 - Unjammed without motion (the gradual release of the object demonstrates

the gradually decreased magnetization)

This demonstrates that the only function is not jamming, and that the motion can be controlled independent of the jamming state. This is described in Lines 490-506.

We acknowledge that we demonstrate this qualitatively, and not quantitatively. We use a magnetic navigation system that is able to create a relatively homogeneous fields and gradients in a certain workspace, where the differences were negligible for the tests we conducted, i.e. a noticeable change in the grip force was not observed while the gripper was translated and/or rotated. Here, we want to highlight that models of magnetic fields can be quite robust, and that they can be utilized to create live control. The location of a structure can be tracked, and the magnetic field magnitude, direction, and gradient at that specific location can be modified based on the models. Therefore, a live control scenario is possible.

- In the distributed jamming example, we show that we can rotate structures while jammed at specific fields. Therefore, we have demonstrated that structures with a tunable mechanical behavior can be also manipulated in space. Here we have shown it interact with a surrounding matrix. We have attached an additional video (RV1) for the reviewer to reference, which demonstrates extensions of this experiments that show how at different fields, different interaction possibilities emerge. It can be observed in this video that at higher fields, the structures are stiffer and can deform the surrounding matrix more, whereas at lower fields, the interactions are gentler. There has been prior work on magnetic assemblies being manipulated with magnetic fields, to achieve specific motions or assembly geometries ⁷⁻⁹. These works utilize homogeneous magnetic particles, which only enable attraction forces in shape-dependent directions. Our work could be incorporated into such systems to add an additional layer of shape-independent, directionally-controllable mechanical tunability.

5. This article claims that the jamming is reversible (page 6 line 101). But all the demonstrations, especially the ones shown in the supplementary videos, do not contain reversible jamming and unjamming. For example, in supplementary video 1, the 1-D, 2-D, and 3-D tests start with an already jammed state and become unjammed when the magnetic field is removed. Will they be jammed again if the magnetic field is applied afterwards? It looks like the jamming state can only be achieved when all the subunits are well aligned to start with. In other words, the subunits may need to be manually placed into a certain pattern before applying a magnetic field to induce jamming. If that is the case, then this jamming-unjamming transition is not reversible and this point should be elaborated in the manuscript.

2.5 We thank the reviewer for this comment, since reversibility is very important for jamming structures for robotics applications. The reviewer is correct that in SV1, we only show the unjamming behavior. SV2, SV5, and SV 6, on the other hand, demonstrates the jamming behavior. SV7 demonstrates the jamming and unjamming behavior repeatedly. Therefore, we have already demonstrated that the jamming and unjamming behavior is reversible. Our additional experiments and supplementary videos (SV8, SV9, and SV10) also further demonstrate this reversibility.

Additionally, the reviewer has pointed out that the subunits “may need to be manually placed in a certain pattern”. This is true, if we want to achieve a deterministic morphology. And this was

indeed the main approach in the paper, since the focus is on creating tunable isotropic/anisotropic mechanical properties of a structure with a given geometry, and not on the generation of magnetic assemblies by applying external magnetic fields. In light of this comment as well as a comment made by Reviewer 1, however, we have decided to include additional demonstrations to highlight the unique potentials untethered jamming introduces, especially in more random and complex interactions with the environment. Since the subunits can be fully independent from each other, the jamming structure does not have to be one single cohesive body, it can be divided and distributed around the environment and foreign objects. Please refer to Figure 5c and Supplementary Video SV10 for the demonstration of distributed jamming. In these videos, it can be seen that the jamming behavior depends on the subunits' relative positions, and the behavior is much more stochastic. Additionally, it can also be seen that the jamming and unjamming transition is reversible.

The reviewer thanks the authors for the clarification. The reason the reviewer asked about this reversibility is that the conceptual schematics and demonstrations shown in Fig. 1 illustrate a non-reversible unjamming process, but the manuscript claims reversibility. The reviewer suggests adding a clear discussion about when the jamming and unjamming is reversible and when it is not. The current Fig. 1 gives the reviewer (and possible other readers) a first impression that this unjamming process is one-off.

2.5. We thank the reviewer for revisiting their original comment about irreversibility, and agreeing that we have demonstrated the reversibility of jamming structures.

The funnel experiments are used in the first figure as they are a canonical and typical demonstration of jamming. They illustrate two main things which are the main contributions of the paper:

- That the jamming in question relies specifically on magnetic fields, since when the magnetic field is removed, the structures fully pass through the funnel. This would not have occurred with granular jamming that occurs due to geometric interlocking
- That the jamming actuation is fully untethered. Any other former actuation modality for jamming would require a wire or a tube.

Here, we also want to highlight that the jamming itself (which is caused by an increased attraction force between neighboring elements) is fully reversible in all conditions, as we have demonstrated with multiple videos and experimental results. Even in the funnel experiment, the pieces that have fallen through the funnel can be re-jammed afterwards, just in a new configuration. The part that is not reversible which the reviewer is referencing to is reassembly of the subunits in to their original configuration. This is a separate and stochastic phenomenon regarding (self)assembly and disassembly, influenced by the distance between multiple neighboring sub-units, which is particularly visible in our distributed jamming demonstration.

It might be helpful to think about an example in between what we have shown in Figure 1 and in Figure 4, where the individual constituents connect to each other with a loose string, shown below in Figure R2.3. Here, the length of this string could be determined such that the individual subunits are close enough to their neighbors even when there are no jamming forces holding them together. This will allow them to reversibly re-assemble into their original arrangement. Since assembly is not the focus of the present manuscript, we have not included such examples.

Figure R3.3 Subunits held together with a loose string, enabling re-assembly.

We have added clarifying language around reversibility in the main manuscript to reflect these definitions better.

6. Page 5 - Line 91-94: *“The effective susceptibility of the subunits (χ_{eff}) can be determined experimentally or numerically by $H = M \chi_{eff}$, where M is the resulting magnetization of the subunit, and H is the applied external field. [14]”* This equation is wrong. It is unclear if this error is just a simple typo, or this wrong equation is used in the development of the analytical model and numerical calculations. Since this mistake is repeated in Methods - Numerical simulations (page 29 line 637), the reviewer is worried that this mistake is probably not just a typo and could have caused a systemic miscalculation in the analytic and numerical models.

2.6 This is indeed a typo in the manuscript, and we thank the reviewer for noticing. The analytic and numerical models have the correct equation in the models, and we have corrected the text in the manuscript.

The reviewer thanks for this clarification.

7. *What are the soft-ferromagnetic materials exactly? It is not mentioned in the Methods section or SI.*

2.7 They are iron based magnetic materials with a magnetically soft material, such as Ni, Fe, and Fe₃O₄. In our case we used 304 stainless steel wires. We have added this information to the methods section.

The reviewer thanks for this clarification.

8. *It lacks obvious applications of the proposed jamming mechanism. Lots of potential applications are discussed in the section of Discussion. But none of these mentioned applications are demonstrated or can be directly inferred intuitively from the demonstrations presented in the current manuscript.*

2.8 We have added two new demonstrations to help connect to potential applications. These are already integrated into the comment 2.3 above, and can be found in the new section in the paper titled Interaction with Objects and Environment.

The reviewer thinks these two added demonstrations are overly simplified in comparison with existing similar demonstrations in the literature, and do not show the benefits or advantages of the newly proposed jamming mechanism.

We appreciate the reviewer's comment. While we agree that the demonstrations in this manuscript are relatively simplified, we would like to reemphasize the present research's main contribution and focus: establishing a new magneto-mechanical jamming phenomenon that enables fully untethered jamming/unjamming and multi-directional control of stiffness and yield. Thus, the demonstrations in this manuscript are designed to showcase remote jamming capability, which benefits from selectively adapting to the environmental conditions. Therefore, the scenarios were intentionally chosen to highlight these core mechanical capabilities, specifically focusing on demonstrating untethered jamming structures interacting with diverse environments (in terms of stiffness and topology) and on utilizing multiple degrees of freedom offered by magnetic field navigation systems.

We have modified the text in the discussion to better reflect this focus.

Reviewer #3

The authors have made sufficient modifications in response to the reviewers' comments. In particular, they have adequately addressed my concerns in the revised manuscript. I have no further comments to improve this work.

3.0 We are pleased to hear that our modifications are sufficient, and that we have addressed concerns raised during the first revision. We would once again like to thank the reviewer for their thoughtful questions and suggestions, we strongly believe their contributions have improved our manuscript's quality and impact.

References

1. Gu, H. *et al.* Magnetic cilia carpets with programmable metachronal waves. *Nat. Commun.* **11**, 2637 (2020).
2. Karacakol, A. C., Alapan, Y., Demir, S. O. & Sitti, M. Data-driven design of shape-programmable magnetic soft materials. *Nat. Commun.* **16**, 2946 (2025).
3. Rafiee, M., Farahani, R. D. & Therriault, D. Multi-Material 3D and 4D Printing: A Survey. *Adv. Sci.* **7**, 1902307 (2020).
4. Mazeeva, A., Masaylo, D., Razumov, N., Konov, G. & Popovich, A. 3D Printing Technologies for Fabrication of Magnetic Materials Based on Metal–Polymer Composites: A Review. *Materials* **16**, 6928 (2023).
5. Demirörs, A. F., Pillai, P. P., Kowalczyk, B. & Grzybowski, B. A. Colloidal assembly directed by virtual magnetic moulds. *Nature* **503**, 99–103 (2013).
6. Martin, J. J., Fiore, B. E. & Erb, R. M. Designing bioinspired composite reinforcement architectures via 3D magnetic printing. *Nat. Commun.* **6**, 8641 (2015).
7. Yigit, B., Alapan, Y. & Sitti, M. Programmable Collective Behavior in Dynamically Self-Assembled Mobile Microrobotic Swarms. *Adv. Sci.* **6**, 1801837 (2019).
8. Yu, J. *et al.* Active generation and magnetic actuation of microrobotic swarms in bio-fluids. *Nat. Commun.* **10**, 5631 (2019).
9. Yu, J., Wang, B., Du, X., Wang, Q. & Zhang, L. Ultra-extensible ribbon-like magnetic microswarm. *Nat. Commun.* **9**, 3260 (2018).

The authors present an untethered jamming mechanism based on magneto-mechanical coupling in soft-ferromagnetic composites. This approach has the potential to address key limitations of existing jamming actuation methods, which often require tethering, making structures challenging to scale and reconfigure. The study establishes design guidelines for programming the magnetization of subunits—the fundamental building blocks of the magneto-mechanical jamming system—to enhance performance. Analytical and numerical studies provide a framework for predicting, programming, and actively controlling the mechanical properties of these magnetic composite subunits. The authors further demonstrate the versatility of the mechanism through multi-dimensional jamming in 1-D, 2-D, and 3-D structures, including the ability to selectively jam along specific directions.

Jamming refers to the phenomenon where an aggregate transitions from a fluid-like state to a solid-like state. While extensively studied, existing jamming actuation methods, such as those using vacuum or voltage, lack untethered tunability and reconfigurability. The main contribution of this study is the introduction of an untethered jamming mechanism enabled by remote magnetic actuation. Overall, this work is both compelling and impactful. The article presents contributions, is well-organized, and mostly well-prepared. I believe the findings certainly deserve publication in *Nature Communications* and will be of considerable interest to researchers in the field of jamming and robotics. I recommend the paper for publication following adequate responses to the comments outlined below.

The attractive force between neighboring subunits plays a pivotal role in the proposed untethered jamming mechanism. Although the experimentally measured attractive force between neighboring subunits agrees with the FEM simulation results in the overall trend (see Fig. 2b), indicating that the jamming force can be remotely tuned, there are still considerable differences in specific values between the experimental tests and the FEM simulation results. The authors might comment on this point. In addition, as mentioned by the authors in the section “Magnetically induced mechanical behavior”, the attraction force between two cubes, if approximated as two soft-magnetic dipoles, can be expressed as $F_{attraction} = \mu_0 H^2 (V \chi_{eff})^2 / (4\pi d^4)$. Can the author use this analytical estimation to predict the attractive force between neighboring subunits and compare the results with the experimental and FEM simulation results?

The reviewer is not clear about how the deviation of magnetization direction (as presented in Fig. 2h-i) is defined. Here free-standing subunits are considered, and an external magnetic field is applied to magnetize them. As far as I understand, the magnetic phase in the subunits will be magnetized along the direction of the applied magnetic field. How is the deviation of magnetization direction formed? Is this deviation angle a volumetrically averaged value?

Regarding the pivoting mode illustrated in Fig. 3f, the subunit will rotate about the pivot point after the breakaway. With larger deformations, the subunit's easy axes will significantly deviate from the direction of the applied magnetic field. In this case, will the subunit be further magnetized along the direction of the applied magnetic field (instead of its easy axis direction)?

Minor points:

In line 303 of page 15, the local internal tensile force should be F_x , and the shear torque should be T_x , according to the notations in Fig. 3h.

The point load in Fig. 3i-j is denoted as F , while in the main text it is represented as P .

In the section “Multi-directional jamming”, the cartesian axes are labeled with lowercase x, y, and z in some places and uppercase X, Y, and Z in other places.

Round 1 reviewer's comment
Authors' responses
Round 2 reviewer's comment

The reviewer thanks the authors for the clarifications, explanations, and added experiments. Unfortunately, the reviewer thinks the critical problems of this work remain.

This article titled “Jamming with magnetic composites” studies an externally controllable and tunable jamming phenomenon of centimeter-scale units. The jamming, or shape-locking, could be achieved along one dimension, in a plane, or in a volume. One difference between the proposed jamming strategy with other existing ones is that it is untethered and wirelessly triggered using a magnetic field. The working principle of this jamming strategy is based on the magnetic properties of soft ferromagnetic particles with low remanence and low coercivity. When an external magnetic field is applied, these magnetic particles are magnetized and they generate attractive forces between each subunit, causing jamming. When the magnetic field is removed, these particles do not retain much magnetization and thus the attractive force disappears, leading to unjamming. This article also characterized the force-deformation behaviors of the jamming structures using analytical models, numerical studies, and experimental validations. The article mentioned that the targeted applications of the proposed jamming mechanism include surgical robotics, micromanipulation, and structural monitoring. The reviewer has the following questions and comments.

1. The main novelty of this work is an untethered jamming mechanism. This article compares the proposed jamming mechanisms with some existing ones, especially the one applies a vacuum to the aggregate with an airtight envelope and the one utilizes a voltage. Then this article claims the key advantage of the proposed one is that it is untethered and easy to scale and reconfigure. However, there lacks a comparison between the proposed jamming mechanism with existing untethered ones. For example, there are previous studies on magnetically induced jamming. Here are some references:

- a) Khalil ur Rehman et al., Magnetic microactuators based on particle jamming, ACS Materials Letters 2024.*
- b) T Leps et al., A low-power, jamming, magnetorheological valve using electropermanent magnets suitable for distributed control in soft robots. 2020.*
- c) Xianhu Liu et al., Magnetic field-driven particle assembly and jamming for bistable memory and response plasticity, Science Advances, 2022.*
- d) Xianhu Liu, Magnetic field-induced particle assembly and jamming, Doctoral thesis, Aalto University, Finland. 2024.*

2.1 We thank the reviewer for their kind reminder and have now cited these references in the revised manuscript. Indeed, these studies are related to magnetic actuation and jamming; however, we would like to highlight that our work is fundamentally different from these in various ways.

- a. All of these references utilize homogeneous spherical magnetic particles. This results in two key differences:
 - i. The homogeneous subunits are not capable of directional controllability of the jamming behavior. They have no pre-programmable easy axes that determine the**

- possible direction(s) of jamming. Our work focuses on the design and creation of specific subunits with the ability to achieve multi-dimensional and programmable directional jamming behavior. Composite design also enables the consideration of the repulsion forces induced between subunits.
- ii. In the jamming of spherical particles, a kinematic deformation mode dominates, where the jamming is a result of the kinematic locking between the constituents based on rearrangement. In all these prior studies, the magnetic attraction between subunits is only a portion of the jamming behavior and it is coupled with the rearrangement of subunits that find energetically favorable arrangements. Our work allows for a focus on the jamming induced purely by magnetic attraction forces.
- b. The jamming mechanism in these prior works operate in a specific confined space that induces varying boundary conditions alongside the magnetic fields, which have an influence on the force chains formed.
 - i. For example, in papers (c and d), the jamming structures are formed on a flat plate, and the magnetic field is applied asymmetrically from underneath this plate. This would induce magnetic field gradients that pull the particles towards the plate, contributing to the jamming force chains. This is actually beneficial for the applications they are concerned with, such as inducing conductive connections between two parallel plates. However, there are also applications in which a jamming structure needs to operate in a free-standing manner. With our experiments such as lifting experiments in Supplementary Video SV2, we look at free-standing jamming structures. The jamming force chains are along a free-standing direction, without the effect of boundary conditions. These demonstrate how the jamming of subunits can be controlled in a fully untethered way, without reliance on boundary conditions.
 - ii. In paper (b), the jamming is utilized for a valve, the subunits are in a confined space within the valve and the electromagnetic actuation mechanism is directly integrated into the valve structure, which needs to be tethered to an external power source.
 - c. And lastly, we kindly disagree with the reviewer's point about paper (a). The work by Rehman *et al.* demonstrates how the bulk magnetic response of a composite (made of magnetic and non-magnetic particles within a porous polymer), could be modified by jamming the aggregate through the swelling of the polymer. The jamming is not induced magnetically, but through swelling. However, we agree that this work is relevant and have referenced it in our discussion. (Lines 647-657)

The reviewer thanks the authors for the explanations.

2. Another key novelty in this article is the ability to selectively jam structures along specific directions. Is it the case that one specific kind of subunits can only be jammed along a predefined direction and cannot be altered after fabrication? It is hinted in the article that the proposed jamming mechanism can overcome the limitation of only having one "easy axis". But based on the presented results, it seems like that one type of subunits can only be jammed along one direction. If another direction is desired, a different type of subunits need to be deployed. It is obvious the jamming direction can be programmed by varying the design parameters. But once the subunits have been fabricated, i.e., all the parameters are fixed, can this jamming direction still be altered on-the-fly?

2.2 The subunits can be jammed along multiple directions after fabrication, this is the main contribution we have highlighted in Figure 4. The proposed 2-D structures can be jammed along two independent axes or both axes simultaneously, **on-the-fly**. Similarly, the 3-D structures can be jammed selectively in all combinations of the three main axes. Figure 4 and Supplementary Videos (SV) 5, 6, and 7 are all demonstrating this. In Fig. 4a and SV5, it can be seen that the same exact subunits in **a 3-by-3 grid can be jammed along x, y, or both dimensions simultaneously**, based on the magnetic field direction applied. In Fig. 4b and SV6, a 3 dimensional structure is demonstrated, where the exact same 6 subunits can be jammed selectively along different dimensions (only x, only y, only z, xy, xz, yz, or xyz), based on the direction of the applied magnetic field. In Fig. 4c and SV7, a 2-D structure is jammed selectively in multiple dimensions. We hope that this clarifies the reviewer's concerns.

The reviewer thanks the author for the response. The reviewer apologizes for not describing this question clearly in the previous round of peer review. The reviewer would like to further explain what the question is. It is stated in the manuscript (both the original and the revised one) that

*“Additionally, the shape-dependent magnetization behavior of soft-ferromagnetic materials was leveraged for designing subunits. Specifically, a continuous soft-magnetic body has **an “easy axis”**, a direction along which it is more energetically favorable to magnetize, typically the more elongated dimension of the body^{15,16}. **If a jamming structure's subunits are made of a soft-ferromagnetic material, jamming will only occur along the longer axis of the subunit and only in one dimension. This limits the possible morphologies and functionalities of the resulting jamming structures.***

*Magnetic composite subunits that have soft-ferromagnetic components within a non-magnetic matrix can help **overcome these limitations. The magnetic constituents' easy axis can be aligned along the desired direction of jamming irrespective of the overall geometry of the subunits, enabling programmability of jamming directions.**”*

This statement identifies the observation that jamming can only occur along one direction (the easy axis) as the limitation and promises the proposed strategy can overcome this limitation and enable the programmability of jamming directions. And the authors are correct that Fig. 4 shows that jamming of a 3-by-3 grid can happen along x, y, or both dimensions at the same time. But the reviewer's concern is that, for one specific subunit, its possible jamming direction(s) is fixed after fabrication. For example, the subunit utilized in Fig. 4 can only be jamming along x, y, or both directions, while the subunit in Fig. 1a can only be jammed along its thickness. The possible jamming directions are always discrete directions, i.e., we get to choose one from a limited number of possible directions. It is not a continuously variable jamming direction. That is why the reviewer suggests to properly define this “programmability”, which is a very broad and strong claim.

3. This article states that an important motivation and application of this jamming mechanism is to assist robots in hard-to-reach space inside human body. But the prototypes presented in the current manuscript are at the scale of tens of centimeters. They are way too large to enter human body. And there aren't any straightforward methods or clear demonstrations that these prototypes could be downscaled by one or two orders of magnitude.

2.3 We understand the concern and appreciate the feedback. We also would like to mention that the response to this question has parallels with the response to point 1.1a above in response to

Reviewer 1, and we have replicated some of the text in case the reviewers don't have access to all responses.

- a. First, we would like to clarify the scale of the experimental data we have provided. The samples are not in tens of centimeters. The subunits sizes range from samples from 3 mm side-length cubes to 8 mm side-length cubes. The 16 mm length cubes were only used for the characterization experiments looking into the different modes of interaction between two components. The goal was to have higher magnetic interaction forces such that we have better signal to noise ratio in our experimental data. The largest multi-subunit jamming structure we have had in the publication has had a dimension of 5 cm. These types of structures would actually get more challenging to actuate if they were at the scale of tens of centimeters, as controlling the magnetic field in such a large space would require magnetic navigation systems much larger than the one which was utilized in this study.

The scale bars in the figures of the manuscript are 0.5 or 1 cm, indicating the reported devices are at least a few centimeters. Based on the demonstrations, at least three subunits are needed for any tasks and the volume taken is significant within a confined workspace.

- b. To address the concern about applicability, we have added more explicit demonstrations illustrating specific functionalities for applications including but not limited to medical robotics. Specifically, in our new section titled "Interaction with Objects and Environment", we have added:
 - i. A magnetic jamming-based gripper that can apply a range of pinching/grasping forces based on the applied magnetic field. (See **Fig. 5a** and Supplementary Video SV8)
The reviewer appreciates the authors' effort to include more application-oriented demonstrations. But the presented gripper demonstration is over-simplified. There are similar (and more capable) demonstrations in the literature (E. Diller and M. Sitti, *Advanced Functional Materials* 2014). This added result doesn't really show what benefits or advantages the proposed strategy has over existing ones.
 - ii. A dilation beam that can be deployed while flexible, and can gradually anchor itself and apply force to soft environments with the application of increasing magnetic fields. (See **Fig. 5b** and Supplementary Video SV9)

In the context of biomedical applications, procedures such as endoscopy or colonoscopy would be a good fit for these new demonstrations. Endoscopes and colonoscopes can have diameters up to 12 mm and, have working channel diameters up to 4 millimeters, which means they can easily deliver tools up to this size. Additionally, wireless capsule endoscopes typically have a size of 11 mm × 26 mm, as described in a review article published in *Nature Communications* in 2024 (1). The types of jamming structures proposed here could either be integrated into these capsule endoscopes or be delivered with endoscopes or colonoscopes. In light of these types of applications, we have chosen to create structures with 3-4 mm sized subunits, which are in a similar size-range to a number of granular jamming studies that also incorporate non-standard subunit designs (2, 3). This size also allows for the easier retrieval of the subunits, as they aren't so small that it becomes challenging to visualize with common medical imaging modalities. We would like to note that these structures are smaller than other magnetically actuated reconfigurable surgical tools in literature such as (4).

The reviewer acknowledges that there are existing magnetically actuated surgical tools at the size scale of millimeter to centimeter. Specifically for the reference mentioned by the authors, i.e., H. Gu et al. *Nature Communications* 2023, this previous work has demonstrated more functionalities such as the integration of flexible PCBs and LEDs, programmable heating surface, integration with capsule endoscope, and integration with a medical grade thoracic catheter. Overall, the previous work presented a comprehensive investigation with its claims (including its size scale) well justified by experimental results. However, this manuscript lacks functionalities other than jamming, and how it could be integrated with medical devices remains unexplored.

- c. The scale we have focused on for this paper is on the order of millimeters, not only due to the application spaces we were interested in, but also to have a design and fabrication platform accessible to researchers at various institutions with diverse sets of resources. However, there are microfabrication methods to scale down these structures to the order of micrometers, utilizing existing microfabrication methods. The 1D structures could be fabricated with a combination of standard photolithography and electroplating. There are studies which have developed fabrication protocols for similar morphologies (5–7). 2D and 3D structures are a bit more complex, Two-photon lithography would be one method to utilize for this purpose, as previously demonstrated in studies such as (8, 9). The reviewer agrees that several technologies exist for downscaling the reported devices. However, it isn't a straightforward path and many times it is even impracticable. This is the reason why some of the most influential work in magnetically controlled small-scale robots are investigations about fabrication, such as Y. Kim et al. *Nature* 2018 and J. Cui et al. *Nature* 2019. As a result, the reviewer can only evaluate the work based on what results have been actually presented, with an extension to what are also reasonably possible but not demonstrated. Unfortunately, downscaling the reported devices is beyond this.

4. An important limitation of the proposed jamming mechanism is that it requires continuous power consumption to maintain the applied external magnetic field. How does the proposed strategy compare with existing jamming mechanisms from the energy consumption perspective? One of the mentioned applications is untethered robots in minimally invasive healthcare, many of which rely on magnetic actuation. How does this jamming mechanism work alongside those magnetically controlled robots? Will they be compatible in the same workspace? Or maybe this jamming mechanism can only work with robots that use a non-magnetic working principle? It is an important point to clarify.

2.4 There are many excellent points within this comment, and we thank the reviewer for bringing all of them up. We have added some high-level comments in the main text, but have a more exhaustive version here:

- a. Regarding energy consumption:
 - i. In the context of medical applications safety considerations are prioritized over energy consumption considerations. In this case, the “default” and “off” state of the mechanism should be designed considering the specific robotic task and context. For example, for devices which can actively apply high-forces to tissues around them at the jammed state, we would want the structure to unjam and soften when there are system-wide issues. In such cases, requiring the jamming mechanism to

only be “stiff” and “high-force” when there is an active magnetic field applied makes more sense. This was our main consideration. However, there could be contexts where a structure would be “safer” when it is in its jammed state, for example when conforming to the tissues around is not wanted, or if the structure is trying to remain anchored in a part of the body against the flow, etc. For structures in such a context, then considering an alternative “default” and “off” state is necessary.

- ii. There are multiple strategies to change what the “default” state might look like, based on the main mechanism shared in this publication:
 - 1) The soft magnetic components used in the composite design can be chosen from a material that has a high-enough magnetic remanence to remain somewhat jammed when an external magnetic field is removed. This material would still have low coercivity, such that it is easy to unjam or jam the structure further.
 - 2) Permanent magnets could be directly embedded into the design of the jamming structure, ensuring a locally applied magnetic field, rather than relying on an external magnetic manipulation system. The jamming behavior can then be further modified with the externally applied magnetic field.
 - 3) Mechanical metamaterial design features could be incorporated into the design itself to induce interlocking behaviors between individual subunits to maintain in the jammed state (e.g. velcro-like surface).

The reviewer thanks the authors for the explanation.

- b. Regarding the point about integrating the jamming structure into existing magnetically actuated systems for minimally invasive medicine, and whether our jamming mechanism is compatible with such systems: This is actually quite straightforward, and was one of the main motivations for the design approach. While this work is mainly about the fundamentals of magnetic jamming, we believe that it is especially relevant and timely for medical applications with the increasing interest and demand in magnetically controlled medical procedures. A number of the clinically-ready magnetic manipulation systems in literature are able to control magnetic field magnitudes, directions, and gradients independently in all three dimensions. All of these enable multiple degrees of freedom. The jamming actuation we have proposed is controlled mainly by **the magnitude of the applied magnetic field and the direction**. The control of typical magnetic medical robots; however, rely on other aspects of the applied magnetic field. While there are many different possibilities, we outline here the most common ones:
 - i. A magnetic field gradient is utilized to create a pulling force in a desired direction. This is a completely different degree of freedom. In this case, the jamming mechanism could be directly incorporated into existing robotic systems without interfering with the actuation based on gradient. (This can be seen in action in our updated manuscript as well, in the new results we have added, specifically with magnetic jamming based gripper demonstration, Figure 5a and Supplementary Video SV8)
 - ii. There are also magnetically actuated robots that are moved through the environment by utilizing a rotational movement (e.g. locomotion through tumbling), by applying a rotating magnetic field. Our 1-D jammed structures would

be very straightforward to integrate into these systems, as they also have a single easy axis which a torque can be applied to through a rotating magnetic field. The ability to rotate jammed systems can also be seen in our new demonstration on distributed jamming. (This is especially visible in Supplementary Video SV10)

There are three most utilized degree-of-freedom of an exerted magnetic field for controlling such devices, i.e., the field strength, the field direction, and the field's variation in space and time. The proposed jamming mechanism requires two of them, i.e., the field direction and the field strength. It is true that we can still use the field's variation in space and time to control other parts independently. However, we should note that a magnetic field's spatiotemporal variance is heavily coupled with the field strength and direction. Whether we use an electromagnetic coil or a permanent magnet to create the magnetic field, we cannot control the field's spatiotemporal variance independently without also affecting its strength and direction. Unless the robot's main and only capability is based on this jamming (such as the gripper demonstration), this jamming mechanism is unlikely to be compatible with other magnetically controlled devices.

5. This article claims that the jamming is reversible (page 6 line 101). But all the demonstrations, especially the ones shown in the supplementary videos, do not contain reversible jamming and unjamming. For example, in supplementary video 1, the 1-D, 2-D, and 3-D tests start with an already jammed state and become unjammed when the magnetic field is removed. Will they be jammed again if the magnetic field is applied afterwards? It looks like the jamming state can only be achieved when all the subunits are well aligned to start with. In other words, the subunits may need to be manually placed into a certain pattern before applying a magnetic field to induce jamming. If that is the case, then this jamming-unjamming transition is not reversible and this point should be elaborated in the manuscript.

2.5 We thank the reviewer for this comment, since reversibility is very important for jamming structures for robotics applications. The reviewer is correct that in SV1, we only show the unjamming behavior. SV2, SV5, and SV 6, on the other hand, demonstrates the jamming behavior. SV7 demonstrates the jamming and unjamming behavior repeatedly. Therefore, we have already demonstrated that **the jamming and unjamming behavior is reversible**. Our additional experiments and supplementary videos (SV8, SV9, and SV10) also further demonstrate this reversibility.

Additionally, the reviewer has pointed out that the subunits “may need to be manually placed in a certain pattern”. **This is true, if we want to achieve a deterministic morphology**. And this was indeed the main approach in the paper, since the focus is on creating tunable isotropic/anisotropic mechanical properties of a structure with a given geometry, and not on the generation of magnetic assemblies by applying external magnetic fields. In light of this comment as well as a comment made by Reviewer 1, however, we have decided to include additional demonstrations to highlight the unique potentials untethered jamming introduces, especially in more random and complex interactions with the environment. Since the subunits can be fully independent from each other, the jamming structure does not have to be one single cohesive body, it can be divided and distributed around the environment and foreign objects. Please refer to Figure 5c and Supplementary Video SV10 for the demonstration of distributed jamming. In these videos, it can be seen that the jamming behavior depends on the subunits' relative

positions, and the behavior is much more stochastic. Additionally, it can also be seen that the jamming and unjamming transition is reversible.

The reviewer thanks the authors for the clarification. The reason the reviewer asked about this reversibility is that the conceptual schematics and demonstrations shown in Fig. 1 illustrate a non-reversible unjamming process, but the manuscript claims reversibility. The reviewer suggests adding a clear discussion about when the jamming and unjamming is reversible and when it is not. The current Fig. 1 gives the reviewer (and possible other readers) a first impression that this unjamming process is one-off.

6. Page 5 - Line 91-94: *“The effective susceptibility of the subunits (χ_{eff}) can be determined experimentally or numerically by $H = M \chi_{eff}$, where M is the resulting magnetization of the subunit, and H is the applied external field. [14]”* This equation is wrong. It is unclear if this error is just a simple typo, or this wrong equation is used in the development of the analytical model and numerical calculations. Since this mistake is repeated in Methods - Numerical simulations (page 29 line 637), the reviewer is worried that this mistake is probably not just a typo and could have caused a systemic miscalculation in the analytic and numerical models.

2.6 This is indeed a typo in the manuscript, and we thank the reviewer for noticing. The analytic and numerical models have the correct equation in the models, and we have corrected the text in the manuscript.

The reviewer thanks for this clarification.

7. *What are the soft-ferromagnetic materials exactly? It is not mentioned in the Methods section or SI.*

2.7 They are iron based magnetic materials with a magnetically soft material, such as Ni, Fe, and Fe_3O_4 . In our case we used 304 stainless steel wires. We have added this information to the methods section.

The reviewer thanks for this clarification.

8. *It lacks obvious applications of the proposed jamming mechanism. Lots of potential applications are discussed in the section of Discussion. But none of these mentioned applications are demonstrated or can be directly inferred intuitively from the demonstrations presented in the current manuscript.*

2.8 We have added two new demonstrations to help connect to potential applications. These are already integrated into the comment 2.3 above, and can be found in the new section in the paper titled Interaction with Objects and Environment.

The reviewer thinks these two added demonstrations are overly simplified in comparison with existing similar demonstrations in the literature, and do not show the benefits or advantages of the newly proposed jamming mechanism.